# Experimental warming accelerates positive soil priming in a temperate grassland ecosystem

Xuanyu Tao[1,2,11], Zhifeng Yang[1,2,11], Jiajie Feng[1,2,11], Siyang Jian[1,2,11], Yunfeng Yang[3] ✉, Colin T. Bates[1,2], Gangsheng Wang[4], Xue Guo[1,2,3], Daliang Ning[1,2], Megan L. Kempher[1,2], Xiao Jun A. Liu[1,2], Yang Ouyang[1,2], Shun Han[1,2], Linwei Wu[1,2], Yufei Zeng[3], Jialiang Kuang[1,2], Ya Zhang[1,2], Xishu Zhou[1,2], Zheng Shi[1,2], Wei Qin[1,2], Jianjun Wang[5], Mary K. Firestone[6,7], James M. Tiedje[8] & Jizhong Zhou[1,2,7,9,10] ✉

Unravelling biosphere feedback mechanisms is crucial for predicting the impacts of global warming. Soil priming, an effect of fresh plant-derived carbon (C) on native soil organic carbon (SOC) decomposition, is a key feedback mechanism that could release large amounts of soil C into the atmosphere. However, the impacts of climate warming on soil priming remain elusive. Here, we show that experimental warming accelerates soil priming by 12.7% in a temperate grassland. Warming alters bacterial communities, with 38% of unique active phylotypes detected under warming. The functional genes essential for soil C decomposition are also stimulated, which could be linked to priming effects. We incorporate lab-derived information into an ecosystem model showing that model parameter uncertainty can be reduced by 32–37%. Model simulations from 2010 to 2016 indicate an increase in soil C decomposition under warming, with a 9.1% rise in priming-induced $CO_2$ emissions. If our findings can be generalized to other ecosystems over an extended period of time, soil priming could play an important role in terrestrial C cycle feedbacks and climate change.

Since the onset of industrialization, global surface temperature has risen considerably due to the accumulation of atmospheric $CO_2$ and other greenhouse gases from fossil fuel combustion and land-use changes, making climate change a major scientific and political issue worldwide[1–3]. As anticipated, the pace of anthropogenic climate change is likely to accelerate, giving rise to more unpredictable and extreme weather patterns[4,5]. To predict and mitigate future climate change, it is crucial to understand the direction, magnitude, and duration of biospheric feedbacks. However, the feedback mechanisms between terrestrial carbon (C) and climate represent one of the largest uncertainties in forecasting future climate warming in Earth system models[6–10]. Despite extensive research over the past three decades,

[1]Institute for Environmental Genomics, University of Oklahoma, Norman, OK 73019, USA. [2]School of Biological Sciences, University of Oklahoma, Norman, OK 73019, USA. [3]State Key Joint Laboratory of Environment Simulation and Pollution Control, School of Environment, Tsinghua University, 100084 Beijing, China. [4]Institute for Water-Carbon Cycles and Carbon Neutrality, and State Key Laboratory of Water Resources Engineering and Management, Wuhan University, 430072 Wuhan, China. [5]State Key Laboratory of Lake Science and Environment, Nanjing Institute of Geography and Limnology, Chinese Academic of Sciences, 210008 Nanjing, China. [6]Department of Environmental Science, Policy, and Management, University of California, Berkeley, Berkeley, California, CA 94720, USA. [7]Earth and Environmental Sciences, Lawrence Berkeley National Laboratory, Berkeley, CA 94720, USA. [8]Center for Microbial Ecology, Michigan State University, East Lansing, MI 48824, USA. [9]School of Civil Engineering and Environmental Sciences, University of Oklahoma, Norman, OK 73019, USA. [10]School of Computer Sciences, University of Oklahoma, Norman, OK 73019, USA. [11]These authors contributed equally: Xuanyu Tao, Zhifeng Yang, Jiajie Feng, Siyang Jian. ✉e-mail: yangyf@tsinghua.edu.cn; jzhou@ou.edu

experimental studies examining the effects of climate warming on various soil C processes have yielded conflicting and controversial results[6,10–16]. This controversy is partially due to the lack of a mechanistic understanding of the interactive processes among plants, soils, and microbes. Soil priming is one of these critical feedback processes, in which fresh plant-derived C input as litter, dead fine roots, and root exudates can alter the decomposition of native soil organic carbon (SOC)[17,18].

Soil organic carbon (SOC) is vital for soil health, food production, ecosystem functionality, and climate regulation[19]. The global SOC pool contains 2,400 to 3,000 petagrams (Pg) of C, more than twice the amount of the atmospheric C pool[20,21]. Soil C stock is ultimately determined by the balance between the C gains from photosynthesis and the C loss through decomposition and erosion[19]. Apart from their direct contribution to photosynthesis, plants can regulate SOC decomposition via soil priming, which is a key driver affecting soil biogeochemical cycles and determining the capacity of soils to function as sources or sinks of atmospheric $CO_2$[22–24]. The input of fresh C from plants can either stimulate (i.e., positive priming) or suppress (i.e., negative priming) the decomposition of the native SOC[25–29], as shown by the observed positive[30–32], negative[33–35], no effects[36,37], or both[38–40]. Although recent modeling analysis have shown that incorporating soil priming into ecosystem models is crucial for predicting global C distributions[41,42], whether, and how, to capture priming effects in SOC models remains an area of exploration[43]. As a result, despite extensive research on soil priming, our understanding of how climate warming impacts on soil priming and the underlying biological mechanisms remains severely limited[35,44,45].

The impact of climate warming on soil priming is highly complex, as it directly or indirectly affects plant and microbial communities, as well as soil biogeochemical processes. These effects largely depend on factors such as plant productivity, microbial community structure and activity, and soil nutrient status. Climate warming can act as a double-edged sword for soil priming: on one hand, it may accelerate priming by stimulating plant growth and biomass[6,7,46] and/or activating microbial groups and functions that facilitate priming;[6,16,46–49] on the other hand, it may impede priming by reducing plant growth and biomass[50] and/or altering microbial community structure and activities in a way that delays priming. In addition, nutrient availability, such as N could be essential in controlling the direction and magnitudes of soil priming[32,48,51] in response to climate warming.

To investigate whether and how climate warming affects soil priming, we added [13]C-labeled straw as plant litter to soils samples collected from a long-term (7 years) experimental warming site in a tallgrass prairie ecosystem in the US Great Plains of Central Oklahoma (34° 59′ N, 97° 31′ W)[52]. Our main scientific questions are: (i) whether and how does experimental warming affect soil priming; (ii) what are the microbial mechanisms underlying soil priming, and (iii) can soil priming and associated microbial mechanisms be incorporated into ecosystem models to improve model performance and reduce model uncertainty? Based on our previous results that experimental warming enhances microbial succession, temporal turnover rates, and potential biotic interactions[14,47,52,53], we hypothesized that warming would accelerate the positive soil priming. Our results reveal that experimental warming indeed strengthens soil priming via activating various functional populations, and that incorporating microbial mechanisms and information into an ecosystem model significantly reduces model uncertainty and improves the accuracy of predictions (Fig. 1).

## Results and discussion

### Long-term warming enhanced the positive priming effect

To determine how experimental warming affects soil priming, eight surface soil samples (0–15 cm) were collected from the warmed (continuous heating at a target of +3 °C above ambient temperature)

and control plots ($n = 4$) in 2016[52,53]. Although the SOC and total nitrogen (N) were not significantly changed by warming during the 7-year experimental warming, the mineral N ($NO_3^-$) was significantly increased in warmed soil plots than control (Supplementary Table 1). Furthermore, the analysis of dissolved organic matter (DOM) composition by Fourier transform ion cyclotron resonance mass spectrometry (FT-ICR MS, Bruker Daltonics, Billerica, MA, USA) revealed that warming increased the relative abundance of tannins ($p < 0.01$, 95% confidence interval) but decreased those of carbohydrates and condensed aromatic compounds ($p < 0.05$, 95% confidence interval) in soil samples (Fig. 2a & Supplementary Fig. 1).

The soil samples were incubated for one week with [13]C-labeled wild oat (*Avena fatua*) straw powder to simulate plant litter decomposition, along with a [12]C-labeled straw addition and no straw treatments serving as isotopic control and background, respectively (Fig. 1). The overall priming effect was positive for the control samples, with $121.1 \pm 10.4$ µg C/g soil (mean ± s.d.) during the 7-day incubation (Fig. 2b and Supplementary Fig. 2). The priming effect in control samples increased the basal respiration by $554 \pm 99\%$, which is within the range as previously reported, up to 1200%[54]. In the warmed samples, warming accelerated the positive priming effect to $135.6 \pm 5.2$ µg C/g soil ($593 \pm 143\%$ to the basal respiration), which was overall significantly higher ($p < 0.001$, permutation ANOVA) than the control samples by $12.7 \pm 9.4\%$ (Fig. 2b and Supplementary Fig. 2, and see Supplementary Note 1 for details). Consistently, microbial respiration of warmed samples also increased by $14.2 \pm 12.8\%$ ($p < 0.01$, permutation ANOVA) than control samples (Fig. 2b and Supplementary Fig. 2). The warming-enhanced positive priming effects were also observed in another 63-day incubation experiment (Supplementary Fig. 3 and see Supplementary Note 1 for details).

### Warming altered active bacterial communities

Based on the phospholipid fatty acid (PLFA) analysis, the biomass ratio of fungi to bacteria is $0.055 \pm 0.036$ for warming plots and $0.043 \pm 0.025$ for control plots[55], suggesting that bacterial biomass is much higher than fungi in our field site. Consequently, in this study, we focused on examining the responses of active bacterial communities to climate warming. To this end, we extracted DNA after a 7-day incubation and employed quantitative stable isotope probing (qSIP) to identify and analyze the active bacterial community (Supplementary Fig. 4)[56]. Warming significantly increased apparent active bacterial abundance by $81 \pm 34\%$ ($p < 0.001$, permutation ANOVA) and total bacterial abundance by $44\% \pm 24\%$ compared to the control ($p < 0.001$, permutation ANOVA; Fig. 2c, Supplementary Fig. 5, and see Supplementary Note 2 for details).

A total of 147 amplicon sequence variants (ASVs) were identified as active C decomposers across all samples (Fig. 2d and see Supplementary Note 3 for details), including well-known C decomposers such as *Burkholderia*, *Sphingomonas*, and *Bacillus*[57–59]. The active bacterial community compositions in warming samples were different from those in control ($p < 0.05$, permutation ANOVA, Fig. 2e). Among the active bacterial community (Fig. 2d), 56 ASVs (60%) were active only in warmed samples, indicating that these active C decomposers (e.g., *Bacillales*) exclusively responded to warming treatment (Fig. 2a and see Supplementary Note 3 for details). To confirm this, these ASVs were also detected in situ by annual surveys (during 2010–2016), and their mean relative abundance increased by 27–205% under warming (Fig. 2f and see Supplementary Note 3 for details). In contrast, warming did not affect the mean relative abundance of the remaining active ASVs (non-warming induced) during 2010–2016 (Supplementary Fig. 6 and see Supplementary Note 3 for details). All of these results demonstrated a substantial taxonomic compositional change induced by warming, which could affect overall soil C decomposition (Supplementary Fig. 7 and see Supplementary Note 3 for details). In addition, warming increased the phylogenetic α-diversity of the active

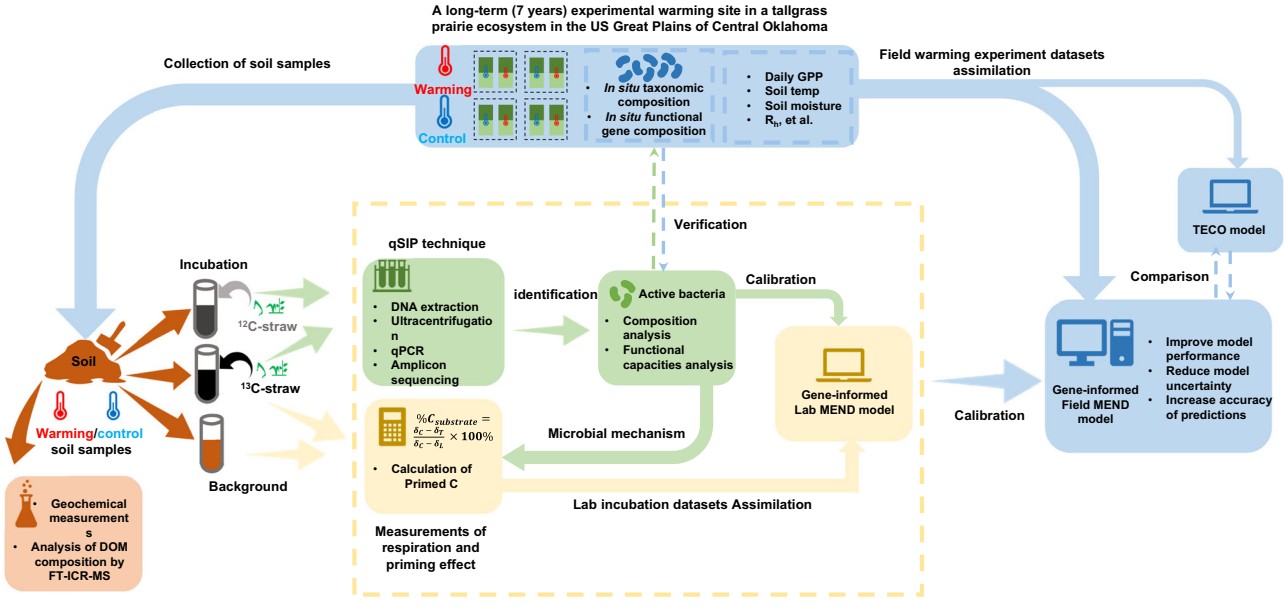

**Fig. 1 | Study design and objectives.** The schematic of the study design illustrates soil sampling from warmed and control plots, followed by comprehensive analyses of soil and microbial properties and mechanisms, and subsequent field model optimization based on lab-derived data/model. The research objectives of this study are to ascertain: (i) the effects of experimental warming on soil priming; (ii) the microbial mechanisms underlying soil priming; and (iii) the potential for incorporating soil priming and associated microbial mechanisms into ecosystem models to enhance model performance and reduce uncertainty. To address these objectives, eight surface soil samples (0–15 cm depth) were collected in 2016 from warmed (targeted continuous heating at +3 °C above ambient temperature) and control plots ($n = 4$) at a long-term (7-year) experimental warming site in a tallgrass prairie ecosystem in the US Great Plains of Central Oklahoma (34°59′N, 97°31′W). Following geochemical measurements and Dissolved Organic Matter (DOM) analysis, the samples were incubated for one week with ¹³C-labeled wild oat (Avena

fatua) straw powder to simulate plant litter decomposition, with additional treatments involving ¹²C-labeled straw and no straw serving as isotopic control and background, respectively. Active degraders in both warming and control samples were identified using qSIP analysis to further explore the microbial mechanisms underlying soil priming. Subsequently, the lab incubation datasets were integrated into a lab-scale Microbial-ENzyme Decomposition (MEND) model to simulate the 7-day incubation period. This lab-MEND model informed the prior parameter range for a separate field-scale MEND (field-MEND) model, which assimilated field warming experiments conducted from 2010 to 2016 to simulate soil C decomposition. Concurrently, the field-MEND model was compared with the Terrestrial ECOsystem (TECO) model to validate the effectiveness of incorporating microbial data into the MEND model for improving performance and reducing uncertainty. Soil, plant-straw, and bacterial symbols, as used in our previous study[55] are adopted here.

bacterial community (Supplementary Fig. 8, Supplementary Fig. 9, and see Supplementary Note 3 for details).

To determine whether and how warming affects the functional capacities of the active microbial communities, GeoChip 5.0 microarray was used. In line with the changes in the taxonomic composition, functional gene composition associated with C decomposition of the active communities under warming was different from those under control conditions ($p < 0.05$, permutation ANOVA, Fig. 3a). Moreover, the taxonomic composition (Fig. 3b, correlation = 0.88, $p < 0.05$, Procrustes analysis) and the DOM composition (Fig. 3c, correlation = 0.78, $p = 0.06$, Procrustes analysis) were significantly or marginally significantly associated with functional gene composition. Fifteen out of 45 detected C-decomposing genes, especially those involved in hemicellulose, cellulose, and lignin decomposition, significantly increased under warming (Fig. 3d). As the second most abundant component of oat straw (~27% w/w)[60], hemicellulose is more chemically labile than cellulose and lignin, making it a preferred substrate for active decomposers during 7-day incubation period. Correspondingly, four biomarker genes for hemicellulose decomposition (i.e., genes encoding xylanase, mannanase, xylose isomerase, and an L-arabinose operon comprised of L-arabinose isomerase, ribulokinase, and L-ribulose-5-phosphate 4-epimerase) were all significantly increased by warming (Fig. 3d). Also, BIOLOG EcoPlates showed that the microbial utilization capacity of xylose in warming samples increased by 77.4% (Supplementary Fig. 10a). Furthermore, the gene *axe* encoding acetyl esterase for cellulose decomposition and the genes encoding phenol oxidase and vanillate O-demethylase for lignin decomposition increased in relative abundance under warming (Fig. 3d). Consistently,

the in situ cellulose decomposition rate determined by litterbag experiment was also increased under warming[14].

To verify functional gene results from the qSIP experiment, we further compared them with the changes of the functional gene composition of the in situ soil microbial communities collected from warmed and control plots during annual surveys during 2010–2016. Consistently, almost all of the genes associated with degrading various soil organic C were significantly increased or unchanged by warming ($p < 0.05$, 95% confidence interval; Supplementary Fig. 10b). In 2016, genes important for degrading starch (e.g., *aceB* encoding malate synthase A, *cda* encoding cyclomatodextrinase, and *amyA* encoding α-amylase), hemicellulose (e.g., genes encoding xylanase and mannanase), and chitin (e.g., gene encoding acetyl glucosaminidase) (Supplementary Fig. 10b) were increased. These results appeared to be consistent with the observed decrease in the relative abundances of carbohydrates and condensed aromatics in warming soil samples (Fig. 2a).

## Priming effect is associated with active bacterial communities under warming

To better understand the potential associations between the positive priming effect and the active bacterial community under warming, we conducted a Projection to Latent Structures (PLS) modeling analysis (Fig. 4a) with the presumed relationships (Supplementary Table 2). The analysis found a notable association between soil temperature and active bacterial communities in terms of their abundance (partial $R^2 = 0.46$, $p < 0.01$, based on the PLS model), phylogenetic diversity (partial $R^2 = 0.30$, $p < 0.01$), and functional genes related to C

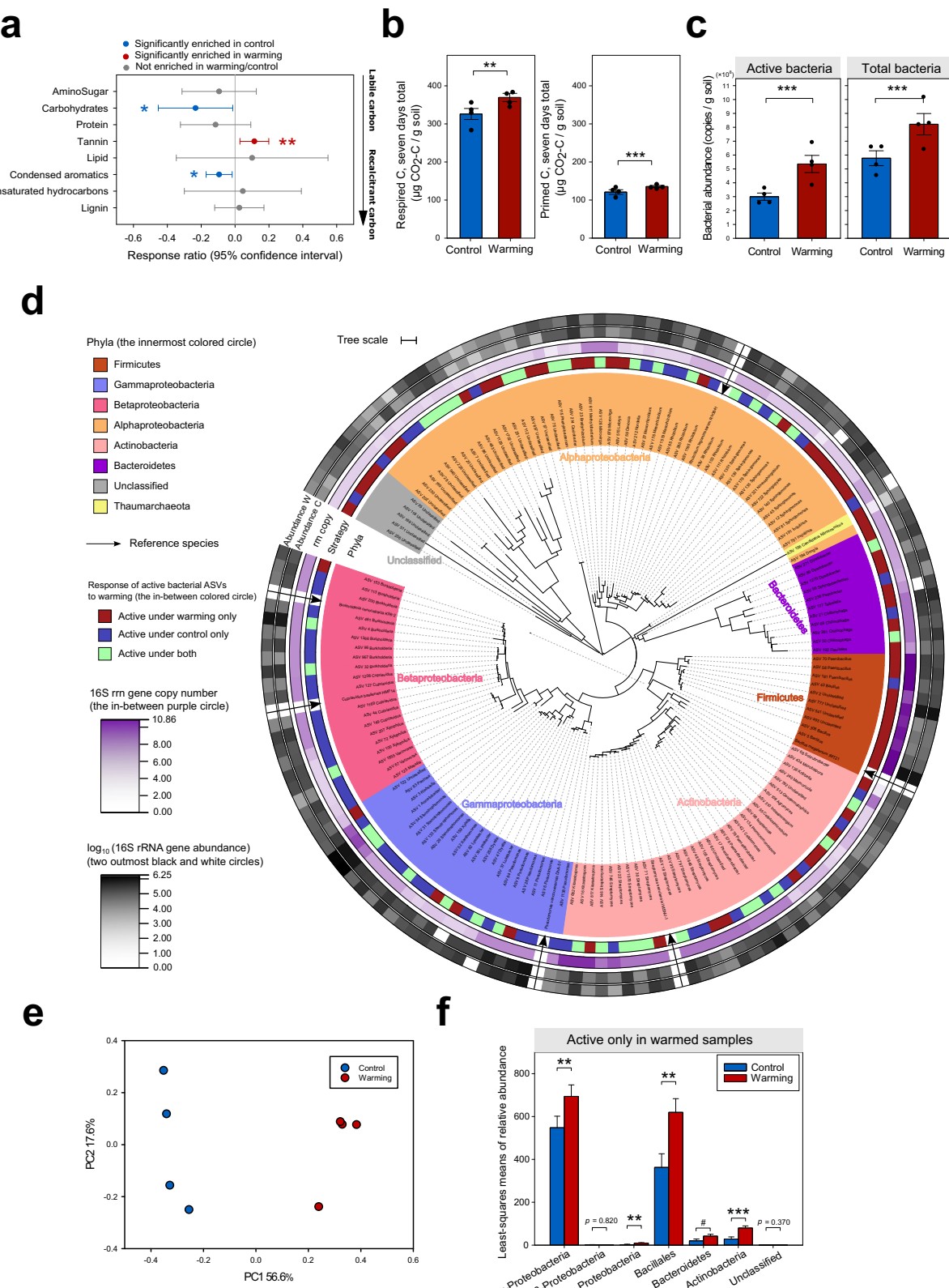

decomposition (partial $R^2 = 0.25$, $p < 0.050$). Mineral N also showed a significant association with soil temperature (partial $R^2 = 0.16$, $p < 0.05$) and active bacterial abundance (partial $R^2 = 0.20$, $p < 0.01$). As anticipated, the composition of active bacterial communities were correlated with total bacterial abundance (partial $R^2 = 0.32$, $p < 0.01$), phylogenetic diversity (partial $R^2 = 0.39$, $p < 0.01$), functional gene abundance (partial $R^2 = 0.47$, $p < 0.01$), carbohydrate (partial $R^2 = 0.34$,

$p < 0.01$), condensed aromatics (partial $R^2 = 0.46$, $p < 0.05$), and tannins abundance (partial $R^2 = 0.34$, $p < 0.05$). Additionally, soil C decomposition had a notable association with active functional genes related to vanillin-lignin decomposition (partial $R^2 = 0.48$, $p < 0.01$), and exhibited a significant correlation with the mineral N (partial $R^2 = 0.42$, $p < 0.05$) in the soil. These findings suggest that priming effects under warming either directly or indirectly related to the

**Fig. 2 | Warming enhanced the priming effect and restructured active bacterial communities. a** Response ratios of relative abundances of DOM between warming and control samples in 2016 (Warming vs. Control). Red symbols indicate significantly positive response ratios, while blue symbols indicate significantly negative response ratios. Grey symbols represent non-significant response ratios. Each symbol represents the average ± 95% CI of four biological replicates ($n = 4$) of warmed or control samples. Significance is denoted as follows: $*p \leq 0.05$ and $**p \leq 0.01$, as determined by using the one-sided Response Ratio test[110]. No adjustments were made for multiple comparisons, and exact $p$-values are provided in the Source Data file. **b** The overall microbial respiration or priming effect during the 7-day incubation with $^{13}$C-labeled straw. The bars represent the average ± standard error of four biological replicates ($n = 4$) of warmed (red) or control (blue) samples. Significance is denoted as follows: $**p \leq 0.01$ and $***p \leq 0.001$ determined by using one-sided permutation ANOVA. Exact $p$-values are provided in the Source Data file. **c** Abundance of active and total bacterial community after the 7-day incubation with plant litter. The bars represent the average ± standard error of four biological replicates ($n = 4$) of warmed (red) or control (blue) samples. Significance is denoted as follows: $***p \leq 0.001$, determined by using one-sided permutation ANOVA. Exact $p$-values are provided in the Source Data file. **d** The maximum-likelihood phylogenetic tree of active bacterial ASVs (decomposers) across all samples. The phyla colors are defined as follows: *Firmicutes* (taupe brown), *Gammarproteobacteria* (lavender purple), *Betaproteobacteria* (pastel pink), *Alphaproteobacteria* (light orange), *Actinobacteria* (dusty pink), *Bacteroidetes* (eggplant purple), Unclassified (light khaki), *Thaumarchaeota* (lime green). W: warmed samples; C: control samples; rrn: 16S rRNA gene. **e** PCoA analysis based on Bray-Curtis dissimilarity metric showing that taxonomic composition of active bacterial communities are different between warmed (red) and control (blue) samples. **f** Yearly means of relative abundance of active bacterial ASVs in in situ warmed samples during 2010–2016. The least-squares mean values were determined by the linear mixed-effects model. Each bar represents the mean ± standard error of 28 biological replicates ($n = 28$) of in situ warmed (red) or control (blue) samples over yearly repeated measures during 2010–2016. Significance is denoted as follows: $\#p \leq 0.1$; $*p \leq 0.05$; $**p \leq 0.01$; and $***p \leq 0.001$, determined by using two-sided ANOVA. No adjustments were made for multiple comparisons, and exact $p$-values are provided in the Source Data file. Source data are provided as a Source Data file.

changes in active microbial community functional genes and N availability (Supplementary Tables 3-6 and see Supplementary Note 4 for details). Moreover, while the inactive bacterial communities were not directly involved in fresh carbon degradation, they could be intricately linked to the consumption of native SOC[61]. Based on Mantel and Pearson correlation analysis, specific taxa within the inactive group (e.g., *Acanthopleuribacteraceae*, *Clostridiales_Incertae Sedis III*, *Chloroflexaceae*, etc.) were found to be associated with the priming effect, suggesting their potential role in the degradation of additional native SOC (Supplementary Table 4, Supplementary Table 5 and see Supplementary Note 4 for details).

Theoretically, two prominent competing hypotheses have been proposed to elucidate the soil priming effect: the stoichiometric decomposition hypothesis and the microbial N mininghypothesis[28,32] (Supplementary Fig. 11). The stoichiometric decomposition hypothesis assumes that microbial activities, including decomposition and respiration, would be highest when the stoichiometry of substrates matches that of microbial demands[32,62] (Supplementary Fig. 11a–d). Specifically, when the C/N ratio exceeds the microbial optimal, an increase in available N draws the C/N ratio closer to this optimal. This enhances microbial activities, thus increasing C decomposition and the priming effect (Supplementary Fig. 11a). Conversely, an increase in fresh C input shifts the C/N ratio away from the optimal, which could induce N deficiency, weaken microbial activity, and lead to a reduction in C decomposition and the priming effect (Supplementary Fig. 11b). In contrast, the microbial N mining hypothesis posits that microorganisms can use labile C as an energy source to decompose the native SOC for additional nutrient/nitrogen (N)[25,63] (Supplementary Fig. 11e-11i).

Our results indicate that (i) Mineral N increased with native soil respiration stimulated by the addition of fresh C (Fig. 4b; $R^2 = 0.51$, $p < 0.050$); (ii) Plant biomass increased with decreased native soil respiration stimulated by the addition of fresh C (Fig. 4c; $R^2 = 0.60$, $p = 0.062$). These observations align more closely with the scenarios predicted by the stoichiometric decomposition hypothesis, especially for the C/N ratio exceeds the microbial optimal scenarios (Supplementary Fig. 11a, b). Furthermore, the stimulation of potential $r$-strategists, such as $\alpha$-*Proteobacteria* and *Bacilli*, by the addition of fresh C (Supplementary Table 3, Supplementary Table 5, and see Supplementary Note 4 for details) is in line with previous observations that r-strategists' dominance elucidates the enhanced SOM decomposition based on the stoichiometric decomposition hypothesis[32]. Therefore, although the roles of microbial N mining could be not completely ruled out, our data are more consistent with the stoichiometric decomposition hypothesis.

## Combination of laboratory and field experiments improved soil C modeling

Since the priming effect has been invoked as an important mediator in regulating the SOC turnover and C cycling and could be enhanced by warming, it is inevitable to be included into global-scale models for better climate projections[43,64]. A great challenge in ecology is to integrate microbial community information, particularly omics data, into ecosystem models[65]. We have recently explored incorporating functional gene data into Microbial-ENzyme Decomposition (MEND)[66]. Since the MEND model considers dynamic microbial dormancy and resuscitation processes regulated by environmental conditions and substrate availability[65,67], it is advantageous to incorporate microbial activity data into ecosystem models. Our previous results indicated that incorporating GeoChip-detected functional genes important to C degradation into the MEND model significantly reduced parameter uncertainties and improved model prediction of soil microbial respiration in response to experimental warming[14], nitrogen amendment, and elevated $CO_2$[66]. While in situ measurements in field experiments is crucial for parameterizing ecosystem models, some parameters (e.g., maximum specific growth rate and dormant microbial maintenance rate) could be efficiently constrained with laboratory experimental observations by excluding various cofounding factors in situ and tracking specific microbial processes[68].

Here, we initially calibrated the lab-scale MEND (lab-MEND) model using laboratory measurements, including microbial respiration (total $CO_2$ and $^{13}CO_2$ fluxes) with or without litter addition, microbial active fraction, and oxidative (EnzCo) and hydrolytic enzyme (EnzCh) concentrations informed by GeoChip-detected functional gene abundances of active communities. Then, we calibrated the field-scale MEND (field-MEND) model using field measurements, including in situ heterotrophic respiration rates ($R_h$), monthly and annual gene abundance, and microbial active fraction as well as lab-MEND derived parameters. For lab-MEND, we found that the simulated total $CO_2$ and $^{13}CO_2$ fluxes agreed well with the observed data from the 7-day laboratory incubation (mean absolute relative error (MARE) = 0.07−0.32; Fig. 5a−c), suggesting that lab-MEND model successfully captured the C dynamics observed by the priming experiments. Additionally, the active fractions based on qSIP and the response ratios of the GeoChip-based functional genes in active communities were assimilated into the lab-MEND model (Supplementary Fig. 14), enabling direct constrains on simulated enzyme production and microbial activation processes.

Since laboratory incubations could overestimate microbial parameters such as intrinsic C use efficiency at reference temperature ($Y_g$)[68], it is necessary to evaluate whether our estimates are reasonable compared to literature before extrapolating them to in situ conditions.

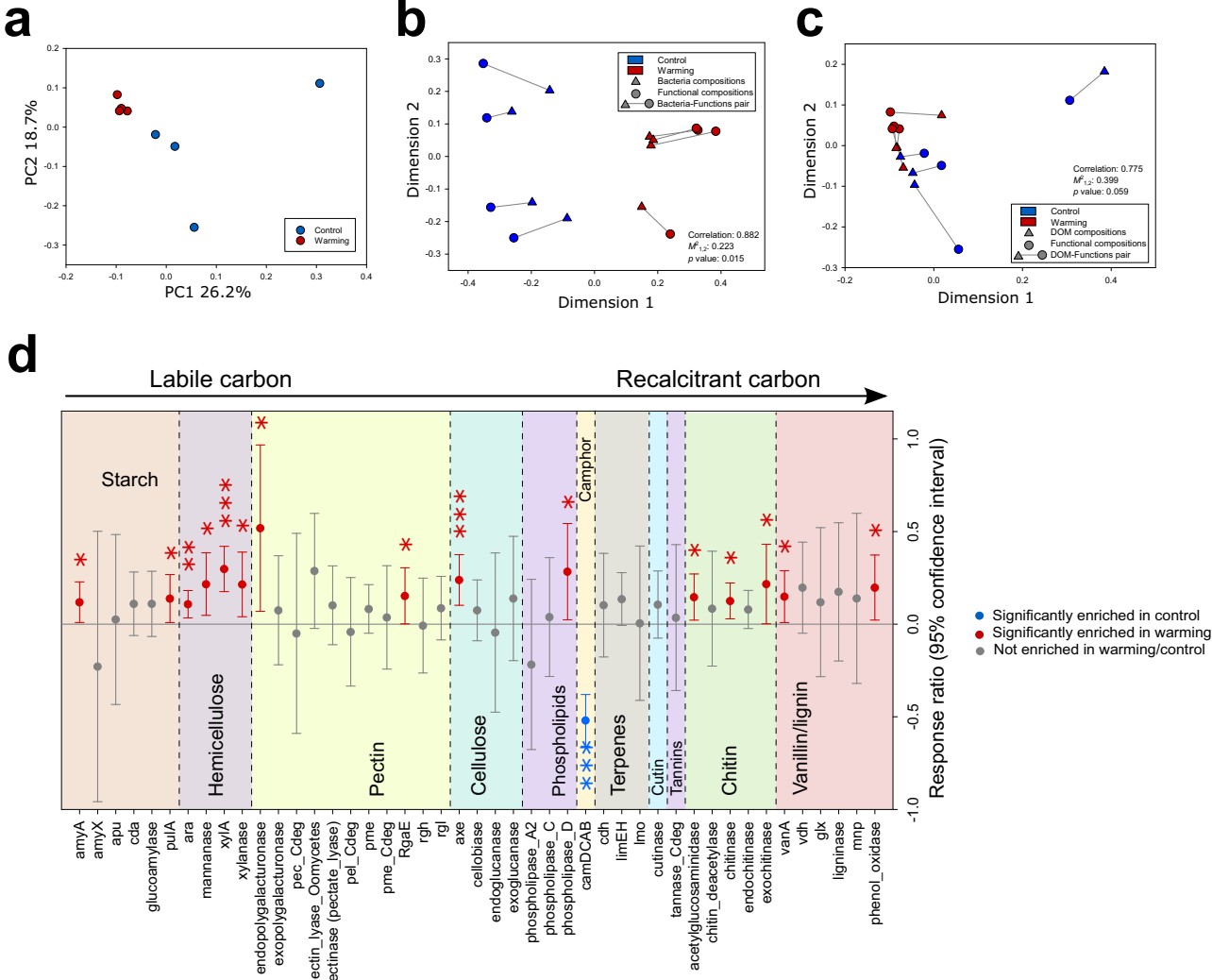

**Fig. 3 | Warming stimulates the C-decomposing capacity and activity of active microbial communities. a** PCoA analysis showing that the functional gene composition of activebacterial communities is significantly different between warmed (red) and control (blue) samples. **b** The congruence between taxonomic (triangle) and functional gene compositions (circle) of active community assessed by a Procrustes analysis optimized through a PCoA plot (red: warming samples; blue: control samples). **c** The congruence between DOM (triangle) and functional gene composition (circle) assessed by a Procrustes analysis optimized through a PCoA plot (red: warming samples; blue: control samples). For (**b**, **c**), The overall fit of the Procrustes transformation is reported as the $M^2_{1,2}$ value. Significance is assessed

using a two-sided PROcrustean Randomization Test (PROTEST)[112], wtih 999 permutations. **d** Response ratios of GeoChip signal intensities of C-decomposing genes between the warming and control samples. Red symbols represent significantly positive response ratios, while blue symbols represent significantly negative response ratios. Grey symbols represent non-significant response ratios. Each symbol represents the average ± 95% CI of four biological replicates ($n = 4$) of warmed or control samples. Significance is denoted as follows: *$p \le 0.05$, **$p \le 0.01$, and ***$p \le 0.001$ as determined by using the one-sided Response Ratio test[110]. No adjustments were made for multiple comparisons, and exact $p$-values are provided in the Source Data file. Source data are provided as a Source Data file.

Our parameter estimates (Supplementary Table 7) are within reasonable ranges compared with previous modeling studies[65,68]. In addition, one key microbial parameter in controlling SOC (i.e., the intrinsic C use efficiency, $Y_g$) was much better constrained when active microbial data were included (0.25–0.32) than excluded (0.47–0.60) ($p < 0.001$; Supplementary Fig. 15a). Also, our estimated $Y_g$ was reasonable compared to global synthesized ranges[69,70].

To determine how microbial genomic information and laboratory data help calibrate ecosystem models, we conducted five model experiments to test different combinations of calibration data for field-MEND and the Terrestrial ECOsystem (TECO) models (Fig. 5d). TECO includes the traditional three soil carbon pools (fast, slow, and passive), but does not explicitly represent microbial pools and their processes[71]. Under both warming and control conditions, TECO had the largest parameter uncertainties on average (Fig. 5d, dark blue bars), potentially due to the lack of microbial and enzyme groups to

assimilate additional microbial information. By contrast, the field-MEND, which assimilated both $R_h$ and in situ gene abundance, reduced the parameter uncertainty by 3–67% (Fig. 5d, yellow bars). The addition of active fraction and lab-MEND-derived parameters reduced the parameter uncertainty by 7–44% and 6-19%, respectively (Fig. 5d, grey and red bars). The field-MEND, which assimilated all information, had the highest performance and reduced the parameter uncertainty by 32–37% compared with field-MEND assimilating $R_h$ and genes ($p < 0.05$, Fig. 5d, light blue bars). In total, the calibration with all data using field-MEND reduced the uncertainty by 78% compared with TECO under warming condition ($p < 0.01$). These results highlighted that integrating diverse microbial data in ecosystem models can effectively reduce parameter uncertainty.

In general, simple models with very few parameters may underfit the experimental data, failing to capture the variations in datasets containing multiple observed variables. Conversely, complex models

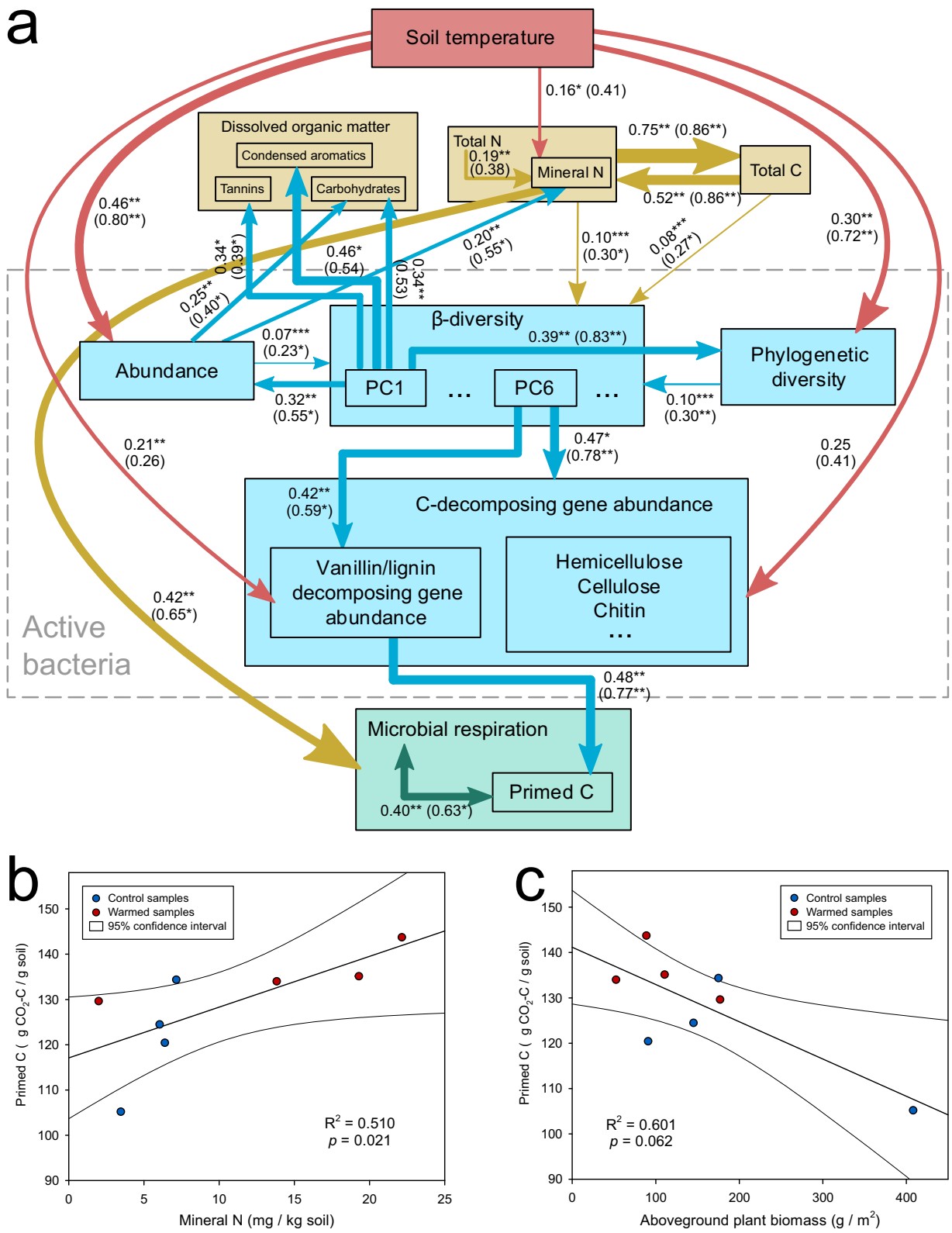

with an excess of parameters may overfit the data, leading to poor generalization[72]. To test the impacts of model complexity on model generalization, we compared the test error (unexplained variance of the test dataset) for field-MEND models with an increasing number of parameters calibrated with the training dataset (Supplementary Fig. 16b and see Supplementary Note 5 for details). Increasing model parameters may lead to reduced unexplained variance in the training set. However, if test error rises simultaneously, it suggests overfitting—the model is overly complex, fitting too closely to the training set and struggling to predict new data accurately. Our results showed that the test error decreased with an increasing number of parameters (Supplementary Fig. 16b), suggesting improved model prediction for test data without overfitting. Nevertheless, in addition to the number of parameters, the consistently higher test error compared to the training

**Fig. 4 | Potential mechanisms of how warming enhances the priming effect.**
**a** PLS model showing the relationships among soil temperature, soil properties, active bacterial community and priming effect. The active bacterial community composition (β-diversity) is represented by the PC1 to PC7 from the PCoA analysis based on Bray-Curtis dissimilarity metric. Directions for all arrows are from independent variable(s) to a dependent variable in the forward selected PLS models ($p < 0.05$ for both $R^2_Y$ and $Q^2_Y$); only the most relevant variables (variable influence on projection > 1) are presented. Each number without parenthesis near the pathway is the PLS partial $R^2$ (Eq. (2)) and the significance is based on permutational test (1000 times) of PLS $R^2_Y$. Each number in the parenthesis is the coefficient of determination ($R^2$) between the two connected variables and the significance is based on Pearson correlation test or Mantel test (for β-diversity). The arrow width is proportional to the strength of the relationship determined by the PLS partial $R^2$. Significance is indicated by *$0.01 < p ≤ 0.05$; **$0.001 < p ≤ 0.01$; and ***$p ≤ 0.001$.
**b** Linear regression between Mineral N ($NH_4^+ + NO_3^-$) from the field and the primed C determined in laboratory (red: warming samples; blue: control samples). **c** Linear regression between plant biomass from the field and the primed C determined in laboratory (red: warming samples; blue: control samples). For (**b** and **c**), p values are calculated using a one-sided permutational test, constrained by treatment and block factors. Source data are provided as a Source Data file.

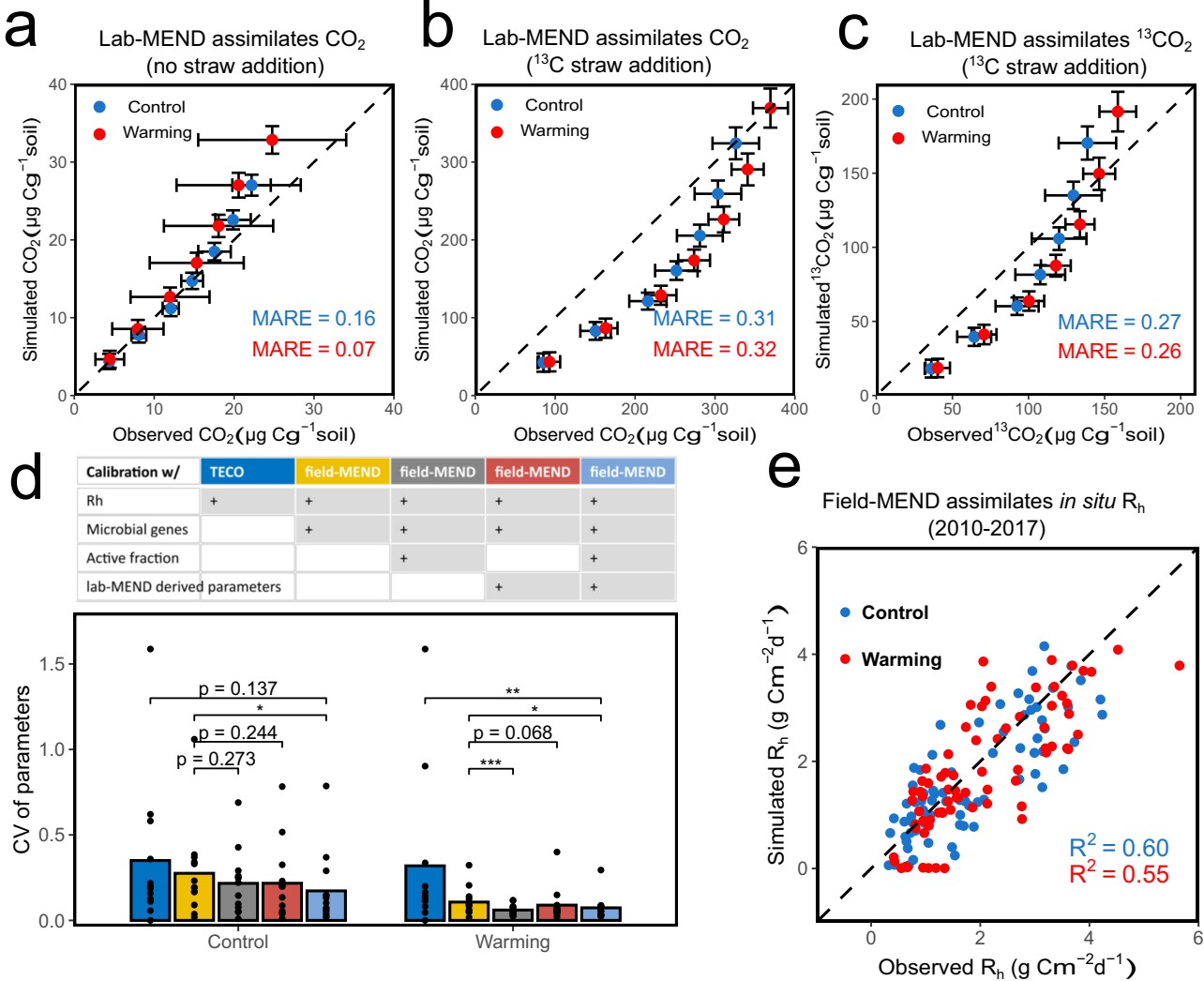

**Fig. 5 | Microbial- ENzyme Decomposition (MEND) model calibration and simulation. a**–**c** Comparison of lab-MEND simulated and observed cumulative microbial respiration in the 7-day qSIP incubation experiment: (**a**) $CO_2$ respired from the soil without straw addition, (**b**) $CO_2$ respired from the soil with straw addition; (**c**) $^{13}CO_2$ respired from the soil with straw addition; For (**a**–**c**), the error bar represents mean ± standard deviation (SD) (red: warming samples; blue: control samples). Observed $CO_2$: mean of 4 biological replicates ($n = 4$); Simulated $CO_2$: mean of 24 hourly simulation values ($n = 24$); MARE: mean absolute relative error (see *Methods* for details). **d** The field-MEND and TECO model parameter uncertainty assessed by the Coefficient of Variation (CV). The bars show the mean CV values of the 13 parameters for field-MEND and 10 parameters for TECO. The bars with different colors represent five model experiments and the above table columns of the same color list the information used in calibration process for each model. The + label indicates usage of the information when doing calibration. Blue: TECO model; Yellow: field MEND calibrated with $R_h$ and microbial genes; Grey: field MEND calibrated with $R_h$, microbial genes and active fractions; Red: field MEND calibrated with $R_h$, microbial genes and Lab-MEND derived parameters; Light blue: field MEND calibrated with $R_h$, microbial genes, active fractions and Lab-MEND derived parameters. Lab-MEND derived parameters indicates replacing the parameter prior ranges with the parameter uncertainty ranges derived from lab-MEND calibration before calibration of field-MEND, which included five parameters (i.e., $p_{EP}$, $V_g$, $\alpha$, $K_D$, and $Y_g$). Significance is indicated by *$0.01 < p ≤ 0.05$; **$0.001 < p ≤ 0.01$; and ***$p ≤ 0.001$ with two-sided Wilcoxon tests. **e**, Comparison of the field-MEND simulated and observed field heterotrophic respiration ($R_h$) under control (blue) and warming (red). $R^2$: coefficient of determination. No adjustments were made for multiple comparisons, and exact $p$-values are provided in the Source Data file. Source data are provided as a Source Data file.

error suggests the need for further enhancements in the model's structure to improve its generalization capabilities[73]. Furthermore, the calibration of more parameters initially increased parameter uncertainty and later decreased it (Supplementary Fig. 16c and see Supplementary Note 5 for details), which indicated that the 14-parameter MEND had the least parameter uncertainty among the models with the lowest test error. Overall, the current parameter selection improves model generalization, alleviating the overfitting problem without a notable increase in parameter uncertainty.

Therefore, we selected the final field-MEND model with the least parameter uncertainties for subsequent analysis. The model simulation achieved a high correlation between the simulated enzyme concentrations and GeoChip-detected oxidative and hydrolytic gene abundances (Pearson correlation $r = 0.59$ for EnzCo; $r = 0.67$ for EnzCh, Supplementary Fig. 14). In addition to the annual GeoChip-detected gene abundances collected from 2010 to 2016[14], the model also successfully fitted the temporal variations of 22 monthly gene abundance data points for 2012 and 2016 (Supplementary Fig. 17). The model simulation also captured magnitudes and changes in microbial active fractions under control and warming conditions ($MARE = 0.12$, Supplementary Fig. 14). These results suggested that the field-MEND model fitted the observed microbial dynamics well with strong constraints applied to microbial processes and their controlling parameters.

We further determined whether the selected field-MEND model has a better predictive capability for $R_h$ compared with TECO (Fig. 5e). The field MEND model showed a high goodness-of-fit between the simulated and observed $R_h$ (Fig. 5e, $R^2 = 0.55$ for warming, $R^2 = 0.60$ for control). In comparison, the TECO model showed a lower performance on simulating $R_h$, specifically for warming conditions ($R^2 = 0.45$ for warming, $R^2 = 0.60$ for control, Supplementary Fig. 14). Hence, this integrated laboratory and field experiments approach with microbial-explicit models improves the accuracy and robustness of representing and modeling microbial feedback in warming and priming processes.

With the calibrated field-MEND model, we further explored the potential contributions of microbial feedbacks to soil $CO_2$ emissions in response to the field experimental warming. Model simulations showed that in situ field warming significantly ($p < 0.001$) increased the 2016 annual average of microbial growth (Supplementary Fig. 15b), the active microbial fraction (Supplementary Fig. 15c), and the decomposition rates of all three soil C pools in the MEND model (Supplementary Fig. 15d). Particularly, the simulated decomposition rate of mineral-associated organic C (MOC, a stable and native C pool with a low reaction rate) increased by 176% under warming (Supplementary Fig. 15d).

Given the enhanced soil C decomposition, it remains an open question whether increased C input to soil in grasslands would result in a net change in soil C, considering both priming and replenishment under warming conditions. In both laboratory and field experiments, the combinations of priming and replenishment yielded net increases in soil carbon (Supplementary Fig. 18) under both warming and control, which is consistent with the results from a previous data-model synthesis study based on 84 priming experiments[74]. Our results also revealed that the experimental warming may decrease the net soil C gain by reducing replenishment by 1.7% (scaled to percent of annual plant C input) and increasing the priming effect by 9.1% under the field condition. Overall, our results suggested that warming enhanced microbial decomposition and priming processes, potentially leading to a reduced net gain from plant C input.

By combining long-term field and laboratory analyses with integrated technologies of isotope chemistry, advanced mass spectrometry, metagenomics, and ecosystem modeling, this study provides several important insights into the impacts of experimental warming on soil priming. First, although soil priming plays a critical role in terrestrial biogeochemical cycling, the direction, magnitude, and drivers of priming within the context of climate change remain elusive[24,43,48]. Second, despite intensive research on soil priming, our understanding of its microbial basis and mechanisms is very limited[75]. Our results revealed that the abundances of the active functional groups critical to soil C degradation were greatly promoted under warming, although the overall diversity of microbial communities is lower under warming[47,53]. This is the first demonstration that experimental warming can regulate soil priming via altering the active microbial community's functional structure. In addition, assimilating various types of field and laboratory data is critical, but very challenging, for global ecosystem modeling[76,77]. Although incorporating functional gene information into ecosystem models is valuable for improving ecosystem model performance as we demonstrated recently[14,66,67], how to effectively use both laboratory and field data to constrain ecosystem models remains challenging. This study represents the first attempt in using both laboratory-derived active microbial data and functional gene information, as well as diverse field observations, for robust model parameterization and simulations. Thus, it is possible to improve the model predictive ability for projecting future climate change by considering various types of feedback mechanisms resulting from both field and laboratory experimental data on plant-soil-microbe interactions and the adaptive changes in active community microbial diversity and structure at the level of functional guilds[6].

Together, our findings have two important implications for projecting ecological consequences of future climate warming and ecosystem management. Firstly, our results revealed that warming intensified positive soil priming effects, leading to increased soil $CO_2$ emissions to the atmosphere, especially if warming coincides with enhanced plant biomass production. Therefore, the effects of climate warming on temperate grassland ecosystems could be more pronounced than previously estimated when the dynamic of soil priming is factored in. This nuanced understanding underscores the complexity of soil carbon responses to climate warming and the need for integrated models that capture the interplay between soil carbon dynamics and ecosystem processes[22]. Second, because the positive soil priming is primarily driven by enhanced microbial activities of several key bacterial functional groups, it could be counteracted for climate change mitigation via in situ engineering of microbiome interactions by adding inhibitive amendments to retard the decomposition activities of key microbial populations[78,79] or by using genetic approaches[80–82] to particularly alter their decomposition activities. However, further research is necessary to determine whether the warming-enhanced positive priming and associated mechanisms learned from this experimental system are applicable to other ecosystems.

## Methods

### Site description and field measurements

The in situ warming experiment was carried out in the tallgrass prairie of Kessler Atmospheric and Ecological Field Station (KAEFS) in McClain County, Oklahoma, USA (34° 58′ 44″ N, 97° 31′ 15″ W). Detailed information of our study site, which was initiated in 2009, is described in our previous study[52]. The soil classification of the site is *Mollisols* (Suborder: *Ustolls*), which is a well-drained soil formed in loamy sediment on flood plains[83]. The soil texture is loam with 51% of sand, 35% of silt, and 13% of clay, with a soil bulk density of 1.2 g cm$^{-3}$. The soil has an available water holding capacity of 37%, neutral pH (6.64 ± 0.24 in 2009), and a deep (about 70 cm), moderately penetrable root zone[52]. The addition of straw powder to soil with a high C/N ratio (C/N ratio: 24), similar as litterfall in natural conditions, could lead to the soil to be more N limited for microbes[84].

Constantan-copper thermocouples wired to a Campbell Scientific CR10X datalogger (Campbell Scientific Inc., Logan, UT, USA) were used to measure and record soil temperature every 15 min at the soil depth

of 7.5, 20, 45, and 75 cm at the center of each plot. In this study, we used the annual average temperature data in 2016 (the sampling year) at a depth of 7.5 cm. Soil volumetric water content at a depth of 0–15 cm was measured every month by placing a portable time-domain reflectometer (Soil Moisture Equipment Corp., Goleta, CA, USA) in three randomly selected locations of each plot. Then the annual average soil moisture in 2016 was calculated. Above-ground plant biomass was measured during the peak growing season (September 2016) by a modified pin-touch method[85].

## Sample preparation and geochemical measurements

Eight soil samples used in this study were collected in September 2016 at a depth of 0–15 cm from 4 warmed plots and 4 control plots (i.e., four biological replicates for warming or control). Visible roots longer than 0.25 cm and stones were removed from soil by 2-mm-mesh metal sieves (Hogentogler Co. Inc., Columbia, MD, USA), followed by manual mixing to homogenize soils thoroughly. All samples were then analyzed for soil geochemistry by the Soil, Water, and Forage Analytical Laboratory at Oklahoma State University (Stillwater, OK, USA) (Supplementary Table 1). Organic C and total nitrogen contents of the soil were determined by a dry combustion C and nitrogen analyzer (LECO Corp., St. Joseph, MI, USA). Soil pH was measured using a water-to-soil mass ratio of 2.5:1 by an Accumet XL15 pH meter with a calibrated combined glass electrode (Accumet Engineering Inc., Westford, MA, USA). The dissolved organic matter (DOM) in both warming and control soil samples was analyzed by an ultrahigh-resolution Fourier transform ion cyclotron resonance mass spectrometer (Tesla solariX XR system, Bruker Daltonics, Billerica, MA, USA), adhering to established methods[86]. Raw spectral data were processed using software BrukerDaltonik v4.2 and Formularity (v1.0)[87]. The identified molecules were subsequently classified into 8 compound classes, based on their respective van Krevelen diagrams[88].

## SIP incubation and measurements of respiration and priming effect

[13]C- and [12]C-straw of common wild oat (*Avena fatua*)[89] was used as stable isotope probe (SIP) substrates to simulate deposition of grass litter to the soil. In general, the change of native SOC decomposition caused by root inputs is referred to as rhizosphere priming effect (RPE), while those by leaf and stem C inputs as litter-derived priming effects (LPE)[26,90]. For the purpose of this study, the priming effects resulting from litter input are referred to soil priming. The C and N contents of the straw are 40.10% ± 0.08% and 1.70% ± 0.05%, respectively, determined by the Soil, Water, and Forage Analytical Laboratory at Oklahoma State University, OK, USA. The [13]C atom% of the [13]C-straw was 75.1%, as determined by the Stable Isotope Facility, University of California, Davis, CA, USA. Before the incubation experiment was set up, the [13]C- and [12]C-straw were ground into powder. Three incubation groups, i.e., (i) with 0.1 g of [13]C-straw in 5 g of soil (equivalent to 10 mg C/g dry soil) as isotopic treatment, (ii) with 0.1 g of [12]C-straw in 5 g of soil (equivalent to 10 mg C/g dry soil) as isotopic control, and (iii) with 5 g of soil as the background, were set up for both in situ warmed and control samples. To homogenize soil samples with straw, these three groups were thoroughly stirred with steel spoons. Each replicate was sealed in a 25-ml lightproof bottle and incubated at 25 °C for 7 days.

Given that cellulose and hemicellulose constitute over 65% of oat straw's composition[60] and hydrolyze faster than lignin, prolonged incubation could lead to severe cross-feeding issues among microbial community members[91,92]. Thus, to minimize the potential for cross-feeding and enhance the accuracy of subsequent quantitative stable isotope probing (qSIP) experiments[58,91–93], the incubation period was set to seven days in this study. This strategy ensures a reliable identification of carbon decomposers. Based on reviewing previous studies which utilized complex carbon (e.g., straws or leaves) to study priming

effects[31,38,94–98], the typical range for C addition amounts is between 2.5 mg C/g soil and 12 mg C/g soil. Additionally, the straw is degraded and assimilated much slower by bacteria compared to glucose, as the turnover time for straw is ~10[4] to 10[6] longer than the glucose[99,100]. A previous study using the qSIP to study the priming used 1.25 mg glucose (equivalent to 0.5 mg C/g soil) as the fresh carbon for 1 g soil incubation[61]. Given these, a relatively higher amount of [13]C-straw is essential for ensuring efficient [13]C incorporation into the DNA of active bacteria and producing a discernible [13]C-incorporation signal for identification of active bacteria during 7-day incubation. Therefore, we decided to add 0.1 g of straw to 5 g of soil (10 mg C /g dry soil) for both determining the priming effect and efficiently labeling DNA of bacteria for qSIP.

However, one week incubation might miss the information of whether more recalcitrant C such as lignin can affect priming. Concurrently, whether smaller straw additions would yield comparable results for the priming effect in response to warming? Thus, considering the potential for continued carbon processing beyond a seven-day incubation period, we also established another 63-day incubation experiment. This incubation experiment was set up exactly as above described, with the only differences being the addition of C at 3 mg C/g soil and an extended incubation period of 63 days.

Headspace gas was collected daily into 12-ml evacuated vials (Labco Ltd., Lampeter, UK), after which the bottles were opened and refreshed for 30 min on a clean bench with the maximal flow of wind. To avoid gas contamination from the atmosphere, we diluted sampled gas by injecting 10 ml of $N_2$ gas into each vial, generating a positive pressure to the atmosphere. $^{12}CO_2$ and $^{13}CO_2$ concentrations were measured at the Stable Isotope Facility, University of California, Davis, California, USA. The concentrations of CO2 and 13CO2, measured in parts per million (ppm), were converted to moles using the ideal gas law ($PV = nRT$)[58]. For this calculation, the pressure ($P$) was set at 101 kPa, and the volume ($V$) was determined by multiplying 25 ml with the $CO_2$ or $^{13}CO_2$ concentration in ppm. The gas constant (R) was taken as 8.314 J K$^{-1}$ mol$^{-1}$, and the temperature ($T$) was maintained at 298 K. Finally, the percentage of the $CO_2$-C deriving from $^{13}C$-straw was calculated as:

$$\%C_{substrate} = \frac{\delta_C - \delta_T}{\delta_C - \delta_L} \times 100\% \tag{1}$$

where $\delta_C$ is the δ[13]C value of respired $CO_2$ from the soil with no straw, $\delta_T$ is the δ[13]C value of respired $CO_2$ from the soil with [13]C-straw, and $\delta_L$ is the δ[13]C value of [13]C-straw.

For all samples, microbial respiration was calculated as the sum of the amount of $^{12}CO_2$ and $^{13}CO_2$. The amount of SOC primed by straw was calculated as total microbial respiration after straw addition minus the amount of C respired from straw, and then minus the amount of C respired from the soil with no straw.

## DNA extraction

To avoid the potential cross-feeding among different microbial populations, we only analyze the active populations in the 7-day incubation experiment instead of the 63-day experiment. After 7-day incubation, soil DNA was extracted with a freeze-grinding method[101], followed by PowerMax Soil DNA Isolation Kit (Cat. No. 12988-10, MO BIO Laboratories, Inc., Carlsbad, CA, USA) according to the manufacturer's protocol. DNA quality was assessed based on spectrometry absorbance at wavelengths of 230 nm, 260 nm, and 280 nm by a NanoDrop ND-1000 Spectrophotometer (Thermo Fisher Scientific, Waltham, MA, USA). The absorbance ratios of 260/280 nm were about 1.8, and 260/230 nm were about 1.7, which were considered good in DNA quality. DNA was quantified by PicoGreen using a FLUOstar OPTIMA fluorescence plate reader (BMG LabTech, Jena, Germany), which showed that DNA concentrations were 49.1 ± 12.7 ng/μl (mean ±

s.d., $n = 24$), with no difference ($p > 0.05$) among warming/control treatments or isotopic treatments. Soil DNA was stored at −80 °C before further analyses.

## Density-gradient ultracentrifugation of soil DNA

To reveal the effect of $^{13}$C-straw incubation on soil DNA density, we performed density-gradient ultracentrifugation[58]. Briefly, we centrifuged 5.1 ml of a solution composed of 3.6 µg of soil DNA (the minimum total DNA amount in all samples), 1.90 g ml$^{-1}$ cesium chloride (Cat. No. 02150589-CF, MP Biomedicals, Santa Ana, CA, USA), and a gradient buffer (1 mM EDTA, 0.1 M KCl, and 0.1 M Tris-HCl), reaching a final density of 1.725 g ml$^{-1}$. The solution was sealed in a polyallomer centrifuge tube (Cat. No. 342412, Beckman Coulter, Brea, CA, USA) with a cordless tube topper and centrifuged on a Vti 65.2 rotor of an Optima L-XP ultracentrifuge (Beckman Coulter, Brea, CA, USA) at 177,000 $g$ and 20 °C for 48 h. The solution from each centrifuged tube was then divided into twenty-four fractions (14 drops per fraction/ ~0.21 ml per fraction). The buoyant density of each fraction was determined by an AR200 digital refractometer (Reichert Inc., Depew, NY, USA). DNA in each fraction was then precipitated with 20 µg of glycogen and two volumes of PEG solution (30% PEG 6000 and 1.6 M NaCl), washed with 70% ethanol, and resuspended in 35 µl of ultrapure water.

## qPCR of 16S rRNA genes

qPCR was used to determine the abundance of 16S rRNA genes in each fraction. Universal primers 515F (5′-GTGCCAGCMGCCGCGGTAA-3′) and 806R (5′-GGACTACHVGGGTWTCTAAT-3′) were used to target the V4 region of 16S rRNA genes[102]. qPCR was performed in triplicate 20-µl reactions containing 10 µl of SsoAdvanced Universal SYBR Green Supermix (Cat. No. 1725274, Bio-Rad, Hercules, CA, USA), 350 nM of each primer, and 1 µl of template, using a thermocycler program of 35 cycles of 95 °C for 20 s, 53 °C for 25 s, and 72 °C for 30 s on an IQ5 Multicolor Real-time PCR Detection System (Bio-Rad, Hercules, CA, USA). Gene abundances (copy numbers) were determined by a standard curve generated with the 16S rRNA gene segment on the TA cloning vector within *E. coli* JM109 cells (Cat. No. A1360, Promega, Madison, WI, USA).

## Amplicon sequencing of 16S rRNA genes

A two-step PCR was performed to generate amplicon libraries of 16S rRNA genes in each fraction[58]. Briefly, the first step of the V4 region of 16S rRNA genes was amplified by the universal primers 515F and 806R in triplicate 25-µl reactions containing 2.5 µl of 10× AccuPrime PCR buffer containing dNTPs (Cat. No. 12339016, Invitrogen, Grand Island, NY, USA), 0.2 µl of AccuPrime High-Fidelity Taq Polymerase, 1 µl of 10 µM forward and reverse primers, and 10 ng of template DNA. The thermocycler program was as follows: 94 °C for 1 min., 10 cycles of 94 °C for 20 s, 53 °C for 25 s and 68 °C for 45 s, followed by a final extension at 68 °C for 10 min. Bead purification was performed to retrieve amplicons generated by the first step, using AMPure XP magnetic particles (Cat. No. A63882, Agencourt Bioscience Corp., Beverly, MA, USA) with a 1:1 volume to the reactions. The second step of PCR also used triplicate 25-µl reactions comprised of 2.5 µl of 10× AccuPrime PCR buffer containing dNTPs, 0.2 µl of AccuPrime High-Fidelity Taq Polymerase, 1 µl of 10 µM 515F and 806R primer combined with the Illumina adaptor sequence (a pad and a linker of two bases, and a unique barcode sequence on the reverse primer), and 15 µl of the purified PCR product of the first step. The thermal cycling condition was the same as the first step except for a cycle number of 20. Triplicate PCR products from the second step were combined, examined for DNA band of 16S rRNA genes by agarose gel electrophoresis, and quantified by PicoGreen.

PCR products from all fractions were pooled at equal molarity and sequenced in the same MiSeq run[103]. First, raw sequence reads underwent PhiX removal, followed by assignment to corresponding samples according to barcodes with 0 mismatches, and trimming of primers using a pipeline built on the Galaxy platform (http://zhoulab5. rccc.ou.edu:8080/). Next, high-resolution amplicon sequence variants (ASVs) with filtered sequencing errors were identified from the reads using the DADA2 procedure[104] with the dada2 package (version 1.12) in R software (version 4.2.2). Given we used the same amount of DNA for each sample, the ASV table was normalized by rarefying the sequence counts of each fraction based on the total 16S rRNA gene copies in the corresponding fraction determined by qPCR. The rrarefy function in the vegan R package (version 2.4.6) was employed for the rarefaction[105]. Lastly, a representative sequence of each ASV was annotated through SILVA ribosomal RNA gene database (v. 132) with a confidence score of 50%[106]. The 16S rRNA gene copy numbers were annotated through the RDP classifier[107].

## Identification and quantification of active bacterial C decomposers

Active C decomposers were identified by $^{13}$C-qSIP[56,108], a newly developed technology with minor modifications. Briefly, the weighted density based on the abundance in each fraction of each ASV (calculated by combining 16S rRNA genes sequencing and qPCR data) was determined for soil samples with $^{13}$C- or $^{12}$C-straw. The density shift (difference of density) of the ASV between $^{12}$C-straw samples and $^{13}$C-straw samples was calculated for all four biological replicates, and a 90% confidence interval (CI) was calculated for the density shift using the bootstrap method with the boot (v.1.3-22) package in R software. The ASV was considered as an active decomposer if the lower bound of the CI was above zero. To determine the total number of 16S rRNA gene copies per gram of soil for each sample (copies/g soil), the total number of 16S rRNA gene copies in each fraction (copies/µl) was scaled by factors such as the resuspension volume, DNA elution volume, the amount of soil used for DNA extraction, and the volume of DNA used for Density-gradient ultracentrifugation, as described by the following equation (Eq. (2)). The relative abundance of each ASV was then calculated by dividing its sequence reads by total sequence reads from above normalized ASV table. From this, the absolute abundance of each ASV in each sample was derived by multiplying the total number of 16S rRNA gene copies by its corresponding relative abundance. This absolute abundance could further be adjusted based on the copy number of the 16S rRNA gene per cell for each ASV. The active bacterial abundance was then computed by summing the absolute abundances of ASVs identified as active carbon decomposers.

$$A_{total} = \frac{\sum_{i=1}^{n}(a_i) \times V_e \times V_r}{V_d \times M} \tag{2}$$

$A_{total,soil}$ is the total copy numbers in soil sample (copies/g soil), $i$ each density fraction, $n$ is the number of fractions, $a$ is the 16S rRNA gene abundance in each density fraction (copies/µl), $V_d$ is the volume of DNA used for Density-gradient ultracentrifugation (µl), $V_e$ is the elution volume of DNA extraction (µl), $V_r$ is the resuspension volume of DNA precipitates (µl), and $M$ is the amount of soil used for DNA extraction (g).

## Determination of functional community structure by GeoChip microarray

The functional capacity of the active bacterial community was determined by GeoChip 5.0S[58]. Briefly, four fractions of each $^{13}$C-straw sample were selected and regarded as representative for the active bacterial community if 16S rRNA genes of the corresponding $^{12}$C-straw samples at the same density fraction were close to zero (Supplementary Fig. 4). Approximately 50 ng of DNA separated from $^{13}$C-fractions in warmed or control samples were amplified using a Templiphi kit (GE Healthcare, Little Chalfont, UK). Then, 2 µg of amplified DNA was

labeled with a fluorescent dye (Cy-3) dUTP using random primers and Klenow fragment of DNA polymerase I at 37 °C for 6 hrs, followed by heating at 95 °C for 3 min. Labeled DNA for each sample was purified with QIAquick PCR purification reagents (Qiagen Inc., Hilden, Germany) and SpinSmart columns (Thomas Scientific Inc., Swedesboro, NJ, USA), dried in a SpeedVac at 45 °C for 45 min., and resuspended in 43.1 μl of hybridization buffer containing 27.5 μl of 2× HI-RPM hybridization buffer, 5.5 μl of 10× CGH blocking agent, 2.4 μl of cot-1 DNA, 2.2 μl of universal standard and 5.5 μl of formamide. DNA was hybridized with GeoChip 5.0 S (60 K) in an SL incubator (Shel Lab, Cornelius, OR, USA) at 67 °C plus 10% formamide and 20 rpm for 24 h. GeoChip arrays were washed and scanned by an MS 200 Microarray Scanner (Roche Inc., Basel, Switzerland) at 532 nm and 635 nm. Raw signal intensities were processed by an online pipeline (http://ieg.ou.edu/microarray/)[83]. In brief, poor-quality spots were first identified and eliminated. This was determined by flagging or if the spots had a signal-to-noise ratio (SNR) less than 2.0. Following this, the normalized intensity for each remaining spot was calculated. This involved dividing the signal intensity of each spot by the total microarray intensity, and then multiplying by a constant value, which is the average signal intensity of all GeoChip data. Finally, a natural logarithmic transformation was applied to the data. The response ratio of signal intensities to warming was calculated as ln ($I_{warming}/I_{control}$), in which $I_{warming}$ is the signal intensity of C-decomposing genes in warmed samples, and $I_{control}$ is the signal intensity of C-decomposing genes in control samples.

## Determination of carbohydrates utilization patterns by Biolog EcoPlates

Biolog EcoPlates (Biolog Inc., Hayward, CA, USA) containing 31 different labile C sources and one control without C source were used to assess carbohydrate utilization capacity of soil microbial community before the SIP incubation. For each soil sample, 0.5 g of soil was mixed with 45 ml of 0.85% NaCl solution, shaken for 20 min at 180 rpm, and settled at 4 °C for 30 min. Subsequently, 1.5 ml of supernatant was mixed with 13.5 ml of distilled water and added onto Biolog EcoPlates with 100 μl of supernatant per well. The Biolog EcoPlates were incubated for 4.5 days using a Biolog Omnilog PM incubator (Torcon Instruments Inc., Torrance, CA, USA) at 25 °C. Color changes of the wells were transmitted to absorbance-time curves. The area under the curves was calculated to assess the utilization of various C sources[109].

## Statistical and phylogenetic analyses

Most statistical analyses were performed in R software (version 4.2.2). The difference among 16S rRNA gene abundances was determined by the one-way ANOVA with a permutation test (Perm-ANOVA) using the lmPerm R package (version 2.1.0). The statistical significance of the response ratio analyses for both the GeoChip and DOM data was assessed using the Response Ratio Test[110]. The structural differences for microbial communities between warming and control samples based on 16S rRNA gene sequence data and GeoChip data were determined by PCoA using vegan R package (version 2.4.6). Since we have identified which ASVs are the potential active degraders in warming or control groups, the unique representative sequences from those active ASVs were used to search for the same ASVs from the 16S rRNA gene sequencing data spanning 2010-2016 (The data is available in the NCBI Sequence Read Archive under project no. PRJNA331185)[14]. The linear mixed-effects model (LMM) in the lme4 R package (version 1.1–35.1) was used to determine warming effects on the relative abundance of those active ASVs during 2010–2016[14]. Based on the LMM, the means of relative abundances for the active ASVs were least-squares means (estimated marginal means) produced by the emmeans function in the emmeans R package (version 1.9.0)[14]. The difference between respiration and the priming effect was determined by repeated-measure ANOVA in the vegan R package (version 2.4.6). Linear models were used

to detect correlations among microbial communities and C fluxes in the stats R package (version 3.5.2), which was subsequently tested for significance by permutation tests in the lmPerm R package (version 2.1.0). Mean values and standard errors of the mean are calculated. Unless otherwise stated, values of $p \le 0.050$ were considered to be significant. The congruence between taxonomic and functional gene compositions, and between the DOM and functional gene compositions was determined by Procrustes analysis[86,111] of PCoA coordinates with the Bray-Curtis dissimilarity metric. The statistical significance of the Procrustes analysis (i.e., M2) was assessed using the PROcrustean Randomization Test (PROTEST)[112]. This involved permutating the data 999 times to evaluate the significance. The analysis was conducted using the vegan R package (version 2.4.6).

The maximum likelihood phylogenetic tree was constructed with the representative sequence for each active ASV. Cultured species of >99.6% 16S rRNA gene nucleotide identity with the 6 top abundant active ASVs was obtained from BLASTn on NCBI (blast.ncbi.nlm.nih.gov/BlastAlign.cgi) and anchored into the tree as reference species. MEGA 6.05[113] was used to construct the phylogenetic tree with MUSCLE alignment, maximum likelihood method, and a bootstrap of 1,000 times. Visualization of the tree was generated by iTOL (itol.embl.de/)[114]. Phylogenetic α-diversity was measured as Allen's phylogenetic entropy, calculated by entropart R package (version 1.6-13)[115]. The phylogenetic groups governed by selection were identified by iCAMP R package (version 1.6.1)[116].

To calculate the relative importance of environmental factors in determining active bacterial phylogenetic diversity, we performed the model selection analysis using the glmulti R package (version 1.0.8)[117]. The importance score of each factor was calculated as the sum of the Akaike weights for all models containing this factor, with a threshold value of 0.8[117].

PLS model was used to explore the relationships among environmental variables, active bacterial communities, microbial respiration and priming effect[118]. Each optimum PLS model is forward selected from all factors which may affect the dependent variable in biology/biogeochemistry, based on predictive performance counting in the explained variation ($R^2_Y$) and model significance (P for $R^2_Y$ and $Q^2_Y < 0.05$, where significant $Q^2_Y$ helps to avoid overfitting). To visualize relevant associations, we only include the most relevant variable(s) with Variable Influence on Projection (VIP) values higher than 1[118]. When used as independent variables in PLS, the active community composition was represented by the PC1 to PC7 from Principal Coordinates Analysis of the Bray–Curtis distance. Inspired by VIP, we proposed a partial $R^2$ index based on PLS to represent the proportion of variance explained by each independent variable (Eq. (3)). As a reference, we also calculated the pairwise correlation coefficient (as well as the $R^2$) among the factors, and the significance is based on Pearson correlation (between vectors) or Mantel test (between distance matrixes). The PLS-related analysis was performed using ropls R package (version 1.34.0)[119], and Mantel test by vegan R package (version 2.4.6)[120].

$$R^2_{PLSj} = R^2_Y \times \frac{\sum_f \left( W^2_{jf} \times SSY_f \right)}{SSY_{cum}} = \frac{\sum_f \left( W^2_{jf} \times SSY_f \right)}{SSY} \tag{3}$$

$R^2_{PLSj}$ Partial $R^2$ of variable j based on PLS.
$W_{jf}$ The PLS weight of variable j on component f.
$SSY_f$ The sum of squares of Y explained by component f.
$SSY_{cum}$ The cumulative sum of squares of Y explained by all components.
$R^2_Y$ The percentage of Y dispersion (i.e., sum of squares) explained by the PLS model.
$SSY$ Y dispersion, i.e., sum of squares of Y.

## Laboratory data assimilation and model simulation

We integrated multiple laboratory-derived datasets (respiration fluxes, SIP-derived microbial active fraction, and GeoChip-detected functional gene abundances) with the Microbial-ENzyme Decomposition (MEND) model (Supplementary Fig. 19)[14,66] and referred it as lab-MEND. MEND describes soil organic C pools, microbial biomass C pools, and enzymes C pools. Plant C first enters three soil organic C pools: two particulate organic carbon (POC) pools and DOC pools. The microbial pool directly takes up C from DOC and produces oxidative (EnzCo) and hydrolytic enzymes (EnzCh) to decompose POC into DOC and mineral-associated organic carbon (MOC). MEND implements the transition of microbes between the two physiological states (active and dormant), which have different C mineralization and enzyme producing abilities. The flux rates are determined by the governing equations (Supplementary Table 8). The allocation of fluxes to different pools are described in Supplementary Table 9. The response functions which scale the flux rates under different pH, temperature and moisture are described in Supplementary Table 10. The definition and initial ranges of parameters used in the equations and functions were described in Supplementary Table 11. MEND was chosen for reproducing C and microbial community patterns in lab and field warming experiments due to its explicit mechanisms explaining priming effects, including stimulated growth of microbial biomass and transition from a dormant to an active state with fresh carbon input.

In each of the control and warming soil, there were 21 respiration data points in total, including 7 $CO_2$ fluxes in no litter treatment, 7 $CO_2$, and 7 $^{13}C$-$CO_2$ fluxes in $^{13}C$ litter addition. Together, there were 42 respiration data points from four cases (2 temperature treatments × 2 litter treatments). Additionally, the active microbial abundance and GeoChip-detected oxidative and hydrolytic gene abundances were quantified in the $^{13}C$-litter addition treatment, providing six more data points for constraining microbial active fraction and oxidative (EnzCo) and hydrolytic enzyme (EnzCh) concentrations as additional objective functions[14].

Five model parameters regulating important microbial traits were calibrated, including microbial growth, maintenance, extracellular enzyme production, and active versus dormant fractions. These microbial parameters were the enzyme production rate ($p_{EP}$), maximum specific growth rate ($V_g$), a ratio ($\alpha=V_{mt}/(V_g+V_{mt})$) relating specific maintenance rate ($V_{mt}$) to $V_g$, half-saturation constant for microbial assimilation of the substrate ($K_D$), and the growth yield at reference temperature ($Y_g$). The multiple-case version of the MEND was used since it allows for using one set of parameters to represent different soils and treatments. We implemented multi-objective calibration of the model[121]. Each objective evaluates the goodness-of-fit of a specific observed variable, e.g., cumulative $CO_2$ efflux, or relative gene abundances (Supplementary Table 12). The parameter optimization aims to minimize the overall objective function ($J$) that is computed as the weighted average of multiple single objectives (Supplementary Table 12)[122].

$$J = \sum_{i=1}^{m} w_i \cdot J_i \quad (4)$$

$$\sum_{i=1}^{m} w_i = 1 \text{ with } w_i \in [0,1] \quad (5)$$

where m denotes the number of objectives, and $w_i$ is the weighting factor for the $i^{th}$ ($i = 1, 2, ..., m$) objective ($J_i$). In the laboratory data assimilation, $J_i$ ($i = 1, 2, 3, 4$) refers to the objective function value for cumulative $CO_2$ efflux, EnzCo, EnzCh, and active fraction, respectively. Since there are far more cumulative $CO_2$ efflux observations (e.g., 42 in

control and warmed soils) than the other variables and $CO_2$ efflux is the most important variable in soil C studies; we assign a much higher weighting factor to cumulative $CO_2$ efflux than the other three objective functions (EnzCo, EnzCh and active fraction), i.e., $w_1 = 5/8$ and $w_2 = w_3 = w_4 = 1/8$.

The mean absolute relative error (MARE) was used to estimate the model performance between simulated and observed cumulative soil $CO_2$ efflux, active bacterial fraction, and functional gene abundances. Calibration was accepted when the MARE was smaller than 0.4[65,68]. We did not adopt coefficient of determination ($R^2$) as the objective function for the respiration fluxes in the laboratory data assimilation and model simulation[58,68] because the $R^2$ would be overestimated for observations that increase steadily such as cumulative $CO_2$ efflux in this case.

$$MARE = \frac{1}{n} \sum_{i=1}^{n} \left| \frac{Y_{sim}(i) - Y_{obs}(i)}{Y_{obs}(i)} \right| \quad (6)$$

MARE represents the average deviations of predictions (Ysim) from their observations (Yobs), and lower MARE values (MARE ≥ 0) are preferred.

The uncertainty of optimized parameters was quantified by the Critical Objective Function Index (COFI) method[65]. The COFI was computed as $J_{cr}$ (Eq. (7)). The feasible parameter space was determined by the parameters resulting in the total objective function values between $J_{opt}$ and $J_{cr}$. The parameter uncertainty ranges quantified in laboratory data assimilation were used in the field data assimilation as described in the next section.

$$J_{cr} = J_{opt} \cdot \left(1 + \frac{p}{n-p} \cdot F_{\alpha,p,n-p}\right) \quad (7)$$

where $J_{opt}$ denotes the minimum objective function value, $n$ represents the number of observations, $p$ represents the number of calibrated parameters, and $F_{\alpha,p,n-p}$ denotes the value of the F-distribution given $\alpha = 0.05$ and the degree of freedom, $p$ and $n-p$.

As $Y_g$ plays a pivotal role in microbial respiration and growth, we conducted a comparative analysis of parameter uncertainty in the model calibration process, with and without the data on active fraction.

## Field data assimilation and model simulation

After calibrating the lab-MEND, we incorporated its parameter ranges, microbial genomic information and field warming experiment datasets into a new MEND model called field-MEND (Supplementary Fig. 20). To determine how microbial genomic information and laboratory data help calibrate ecosystem models, we conducted five experiments with various combinations of calibration data for field-MEND and Terrestrial Ecosystem (TECO) models: i). TECO model calibrated with $R_h$. ii). Field-MEND model calibrated with $R_h$ and microbial genomic information, that is, gene abundances of oxidative (EnzCo) and hydrolytic enzymes (EnzCh). iii). Field-MEND calibrated with $R_h$, genomic information and microbial active fraction. iv). Field-MEND calibrated with $R_h$, genomic information and lab-MEND derived parameter uncertainty ranges. v). Field-MEND calibrated with all available information (Fig. 5d).

To assess the impact of model complexity on generalization and parameter uncertainty, we first divided the dataset into a training dataset (covering the first 3/4 data) for calibration and a test dataset (comprising the subsequent 1/4 data) for evaluating the model's generalization capability. We then calibrated models with varying numbers of undetermined parameters using the training dataset. The parameters selected for calibration at each

level of complexity were determined based on their sensitivities, which were evaluated using the Multi-Objective Parameter Sensitivity Analysis (MOPSA) method (see Supplementary Note 5 for details)[68,123]. For each level of complexity in the MEND model, the uncertainty of each parameter was estimated using the UQ-COFI method[67] and then averaged across all calibrated parameters. By executing the calibrated models, we calculated the goodness-of-fit ($R^2$) for heterotrophic respiration ($R_h$) in the test dataset and compared the unexplained variance ($1 - R^2$) across different levels of model complexity.

TECO is a conventional CENTURY-like model, where the decomposed soil organic C is proportional to the C pool size (Supplementary Fig. 19). In TECO, Plant biomass enters litter C, which is decomposed into $CO_2$ and transformed into soil C pools. Soil C pools are then decomposed into $CO_2$ and transformed into more recalcitrant C. Soil carbon decomposition rates vary with temperature and moisture and are calculated as the product of baseline turnover rate and scaling functions for temperature and water stress. TECO includes the traditional three soil C pools (fast, slow, passive) and can predict ecosystem dynamics. Compared to MEND, TECO does not include specific microbial pools and enzymes, therefore, limits its ability in assimilating microbial trait information[71].

The field warming experiment datasets (e.g., daily GPP, soil temperature, soil moisture, and $R_h$) used for field data assimilation were reported in our previous study[14]. In addition to the annual GeoChip-detected gene abundances collected from 2010-2016, we added 22 monthly GeoChip-detected gene abundances data points collected in 2012 and 2016. As the field dataset has a larger sample size of observation than the laboratory dataset, we could calibrate more parameters for field-MEND. Thirteen model parameters that regulate microbial processes were selected, including three parameters relevant to enzyme production and turnover ($r_E$, $p_{EP}$ and $fp_{EM}$), two parameters relevant to C flow to dissolved organic C ($f_D$ and $g_D$), eight parameters relevant to microbial growth, maintenance and dormancy ($V_g$, $\alpha$, $K_D$, $Y_g$, $k_{Yg}$, $\beta$ and $\psi_{A2D}$) and temperature sensitivity ($Q_{10}$)[124]. We replaced the initial parameter ranges of field-MEND with the calibrated parameter uncertainty from lab-MEND (Supplementary Table 7) before starting the calibration process for the 4th and 5th model experiments. For TECO model, we calibrated 10 parameters, including seven for baseline turnover rates of seven plant biomass, litter and soil C pools as well as three for scaling functions controlling temperature sensitivity and moisture response in C decomposition.

The model parameters of the field-MEND and TECO were optimized by achieving high goodness-of-fit of model simulations against in situ observations. For each of the observed variables, an objective function was assigned to quantify the model performance between observation and simulation (Supplementary Table 12). The parameter optimization aimed to minimize the overall objective function as described in Eqs. 4–5. For this field data assimilation, the coefficient of determination ($R^2$, Eq. (8)) was used to evaluate the $R_h$ because it is frequently measured, and the absolute values can be directly compared between observations and simulations. The *MARE* (Eq. (6)) was used to evaluate the variables (e.g., microbial biomass, active fraction, and the warming response of enzyme concentration) with only a few measurements, and the absolute values can be directly compared. To capture the temporal variations of enzyme concentrations, the correlation coefficient ($r$, Eq. (9)) was used to evaluate the correlation between observed and simulated values, which could omit the magnitude and unit differences in gene abundance data and simulated enzyme concentration. Similar to the laboratory data assimilation, a much higher weighting factor to $R_h$ ($w_1 = 5/9$) was assigned than the other four objective functions (MBC, active fraction, EnzCo and EnzCh, $w_2 = w_3 = w_4 = w_5 = 1/9$). This approach ensures that the model's performance in predicting

$R_h$ is prioritized, while still considering the other variables.

$$R^2 = 1 - \frac{\sum_{i=1}^{n}\left[Y_{sim}(i) - Y_{obs}(i)\right]^2}{\sum_{i=1}^{n}\left[Y_{obs}(i) - \bar{Y}_{obs}\right]^2} \quad (8)$$

$$r = \frac{\sum_{i=1}^{n}\left[Y_{obs}(i) - \bar{Y}_{obs}(i)\right] \cdot \left[Y_{sim}(i) - \bar{Y}_{sim}(i)\right]}{\sqrt{\sum_{i=1}^{n}\left[Y_{obs}(i) - \bar{Y}_{obs}\right]^2}\sqrt{\sum_{i=1}^{n}\left[Y_{sim}(i) - \bar{Y}_{sim}\right]^2}} \quad (9)$$

The $R^2$ value quantifies the proportion of the variance in the response variables that can be predicted from the independent variables. A higher $R^2$ value ($R^2 \leq 1$) indicates better model performance. The r value quantifies whether two variables change in the similar direction and degrees, and higher $r$ values ($|r| \leq 1$) means better model performance.

The field model data assimilation generated five sets of best-fit parameters for the five model experiments. Parameter uncertainty ranges for each set were calculated according to Eq. (7). To exclude differences in mean values of parameters, the coefficient of variation (CV) of parameters was calculated to compare parameter uncertainty between field-MEND and TECO models. Since the field-MEND model calibrated with all available information minimized the parameter uncertainty, we selected it to conduct subsequent model simulation from 2010 to 2016 to evaluate prediction accuracy and explore the potential contribution of microbial communities to in situ soil C dynamics and priming effects.

## Simulation of MEND and TECO for prediction

We ran the model simulation from 2010 to 2016 with the best parameter set obtained from the calibration processes for field-MEND. The simulated values of $R_h$, microbial enzyme concentration (EnzCo and EnzCh) and active fraction were extracted for the measurement periods and compared with the observations to assess model prediction accuracy. We also ran model simulation for TECO and compared its prediction accuracy of $R_h$ with field-MEND. Using field-MEND, we further obtained the simulated microbial growth rate, microbial active fraction, and the decomposition rates of three SOC pools in 2016. Similar to a previous study[74], we conducted both lab-MEND and field-MEND simulations to estimate SOC changes in response to litter/plant addition under control and warming. In plant C-added scenarios, $^{13}C$ labeled plant C and $^{12}C$ labeled native soil C were used to trace $CO_2$ sources. The plant C-added scenario of lab-MEND simulations follows the incubation experiment in this study. In field simulations, plant C input remained constant for the plant C addition scenario, while it was excluded in 2016 for the non-addition scenario. Then, we calculated the annual replenishment (remaining added litter C after microbial respiration), primed soil C (respired soil C with plant C addition minus respired soil C without plant C addition), and the net effect of plant C addition (i.e., the sum of replenishment and priming effect).

## Reporting summary

Further information on research design is available in the Nature Portfolio Reporting Summary linked to this article.

## Data availability

Raw sequences of 16S rRNA gene amplicons after the 7-day incubation with straw are available in the NCBI SRA database under accession number PRJNA595391. Raw sequences of 16S rRNA gene amplicons for yearly warming field sites are available in the NCBI SRA database under accession number PRJNA331185. GeoChip raw and normalized signal intensities are deposited in the BioStudies under accession number E-MTAB-13326. All other relevant data are available in Supplementary Information. Source data are provided with this paper.

## Code availability

MEND model codes are available in Zenodo with the DOI identifier at https://doi.org/10.5281/zenodo.10498280.

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

## Acknowledgements

This work was supported by the National Natural Science Foundation of China grant (41825016/32161123002), Second Tibetan Plateau Scientific Expedition and Research (STEP) program (2019QZKK0503), and the Hainan Institute of National Park grant (KY-23ZK01) to Y.Y., the U.S. Department of Energy Office of Science Office of Biological and Environmental Research Genomic Science program (DE-SC0004601, DE-SC0010715, DE-SC0020163, and DE-SC0010570) to J.Z. and M.K.F., and the Office of the Vice President for Research at the University of Oklahoma to J.Z. The experimental analysis by J.K. was supported by DE-SC0016247. The data analysis performed by D.N. was also partially supported by NSF Grants EF-2025558 and DEB-2129235.

## Author contributions

J.Z. and Y.Y. developed the original concepts. X.T., J.F., and X.Z. designed and carried out the priming, and qSIP experiments. X.T. and J.F. performed the GeoChip hybridization experiment for active communities. X.G. performed the Biolog EcoPlate experiment. J.W. measured the DOM in the soil samples. X.T., J.F., D.N., X.L., S.H., and O.Y. analyzed experimental data and carried out statistical analyses. G.W. developed the MEND model. G.W. and Z.S. instructed the modeling. S.J.

and Z.Y. conducted the MEND modeling work. J.Z, X.T., Z.Y., J.F., S.J., and Y.Y. wrote the paper. X.G., C.B., M.L.K., J.K., L.W., G.W., Y.Z., X.J.A.L., Y.O., Z.F., W.Q., M.F., and J.T. edited the manuscript. Considering their contributions in terms of site management, data collection, analyses and/or integration, X.T., J.F., S.J., and Z.Y. were listed as co-first authors. All authors were given the opportunity to review the results and comment on the manuscript.

## Competing interests

The authors declare no competing interests.
