## [Peer Review File · Nature Communications]

Reviewers' Comments:

Reviewer #1:

Remarks to the Author:

This is an elegant study with noteworthy results, particularly the exploration of microbes involved in the priming effect using qSIP, testing the priming effect in a long-term warming study, and using qSIP data to improve modeling the soil C cycle.

I believe the work will be of some significance to the field, but my own recommendation would be to develop the manuscript for a soil or modeling journal, as I feel the largest contributions this paper could make are in advancing the thinking within the field rather than trying to cast the results as significant for long-term soil C projections. The main issues I see are

1) The manuscript overstates the implications of the one-week study of disturbed soils in jars in the lab for soil carbon loss over decades to centuries. For example, the abstract states "warming-enhanced soil priming can lead to considerably more (~52%) soil C loss by the end of this century... soil priming would considerably strengthen the positive feedbacks between the terrestrial C cycle and climate warming." There are many problems with this extrapolation. First, in general, priming is associated with net soil carbon accrual. See

<https://www.nature.com/articles/s41467-018-05667-7> for a meta-analysis that shows that the fresh plant inputs are larger than the native C loss, so net soil C accrues. Second, native soil carbon in priming studies is operationally defined as any soil carbon that is not the newly-added carbon that elicits the effect. Yet there is a tendency to assume that this native soil carbon is quite old and thus important for long-term storage. The quoted statement in the abstract implies exactly that, but there is no evidence presented here for disproportionate loss of old carbon. This is true of virtually all priming studies. One limitation of that design element is that extrapolating to modeled long-term soil carbon loss is highly speculative. The soil carbon results from the warming field site study show no indication of the expected decline, but rather a tendency (i.e., not significant) toward a +10% soil C increment caused by 7 years of warming.

2) The use of ¹³C-labeled plant material for qSIP allows focus only on those microorganisms that use the ¹³C labeled material to grow, but not on the organisms that specifically are involved in priming, which as the authors note is "an effect of fresh plant-derived C on native SOC decomposition." But the decomposition of native SOM is where the priming action is. The approach used here does not interrogate priming, per se, the mystery of which is about which organisms increase the oxidation of native SOM. Much of the priming literature invokes the idea that they are not, that priming involves the initial response of "r-selected" taxa (or copiotrophs, depending on the particular argument) that then trigger the response of "K-selected" taxa (or oligotrophs). The approach used here only identifies one group and not the group that is particularly interesting because of their possibly important biogeochemical role. (It is possible to distinguish these groups using qSIP. See <https://doi.org/10.1038/ismej.2017.43>.) For this reason, this current study is not particularly informative about the microbiology of the priming effect.

As an example of how microbial data can be used to improve soil C models, this work is strong. I'd like to see this more developed, more clearly explained, the sensitivities thoroughly explored, and alternative model formulations formally compared. As it is, while the optimization shows an improved fit using the approach described, there is clearly room to develop this further. Some still argue that classic 3-pool carbon models are "better" in the sense that they are simple and not overly parameterized. Microbial ecological data can rigorously improve this if the data represent actual processes that occur in soil that are represented clearly both in the data and in the model. I can't tell how close this study comes to meeting that high bar, because the model is not sufficiently described nor how the data are treated to compare with model output. These authors could build a stronger paper largely around this point in my view.

Reviewer #2:

Remarks to the Author:

The manuscript explored the litter-induced priming effect based on a 7-day incubation experiment with ¹³C labeled straw addition under control and warming treatments. The research theme is

important and the scope fits the journal. The method is sound and relevant data analysis and modeling approach are appealing. Though the experiment apparently involves two factors, i.e., temperature and straw addition, the current focus on priming, partly use of the entire data, only discussed the straw-induced respiration change under different temperatures. My first concern lies in the lack of details and clarifications of the experimental design and its associated research objective. Whether this experiment is designated for a priming study or factorial study, justifications are needed simply because your research objective, hypothesis, and experimental design remain essentially and coherently consistent or support each other. This ambiguity is expected for clearance in the revised version. The work showed superior strength in the intention of employing various analyses but their connectivity and crossing support seem poor. Overall, the manuscript is potentially publishable after some major and minor revisions. Line comments were also listed below.

Lines 34-37. Too long and too vague. Short sentences are recommended.

Lines 41-42. Do you mean model results?

Line 88. Do you mean microbial-based mechanisms?

Line 94. Do you mean the model already has incorporated soil priming effect and because of the warming-enhanced soil priming effect, different simulation results reached as compared with the model excluding soil priming. The confusion lies in the way how soil priming was included in your model.

Line 374. reference is missing.

Lines 398-400. Is this the definition of priming effect? is it the same calculation in control and warming treatments? did you report the respiration from straw and how does it differ between control and warming treatments? It appeared that a report of respiration in different treatments would help given that the experiment is basically a two factor experiment involving temperature and straw addition. Priming as named is just a derived thing from the experiment. The relationship of the entirety experiment and specific focus on priming seems vague in the current form.

Reviewer #3:

Remarks to the Author:

Disclaimer: I was asked to specifically comment on the soil microbiota characterisation using high-throughput sequencing in this study. While reviewing I have also identified other points that I have listed below.

Methods:

L323: Port-Pulaski-Keokuk complex is not a soil type or at least none in a commonly used classification system. Please provide the soil type according to USDA or FAO/WRB.

L326: I am surprised by this statement as a coarse soil like the one presented here (close to sandy loam in the texture triangle) is usually considered to have a medium to low water holding capacity. Also please provide the value for pH here. Later on the authors describe the measurement of pH so I wonder why this information was provided twice?

L327: please provide more information to the statement of limitation. How low are the C/N values and who is actually limited by C and N (I guess soil microbes?). I am not sure soil itself can be viewed as limited by C and N.

L360: I am not an expert on priming but does priming induced by milled and added litter represent priming after all in terms of magnitude? If not, how can the authors be convinced that their upscaling is useful?

L364: The amount of straw added represents 2% C per g soil (unclear if fresh or dry weight). What was the original C content and how much C (in % to soil dry weight) did the soil used for incubations contain?

L374: Reference missing as indicated by '(ref)'. Also I don't understand the argument here. The authors argue that fermentation products from switchgrass can be produced by pure cultures in 5

days. The authors want to prevent cross-feeding and keep the incubation for 7 days. Does that mean that the authors assume the decomposition takes longer in the soil setting as compared to the pure culture study? I believe that 7 days can still be considered to be fine but the argument above was not clear to me.

L381: Why are these results presented in the Methods section?

L412: I am surprised to see that the DNA concentrations were within a range of 2.6 ng/ μ L (stddev or err?) across all samples. I wish I would have ever been able to extract DNA from soil in such a reproducible manner.

L431: Please provide references for the primers used (Caporaso et al. 2011?).

L464: it does not become clear to me what the "random resampling" approach means and unfortunately the reference provided here does not give more detailed information (it is pretty much the same sentence there). Did you multiply the relative abundance of each ASV with the gene copies per sample? Please clarify and modify in the manuscript.

L525: where does this data come from? It seems it is not described in the presented manuscript as amplicon sequencing is only described for the qSIP fractions. How would the authors infer activity of bacteria from standard MiSeq data?

Additional comments:

L122-126: The reference provided is not complete in the bibliography. Also I find this statement a bit misleading as it leaves out fungi completely which are commonly known to be very important decomposers of soil litter and they should be rather abundant in grassland soils as well (AMF, saprotrophs, etc.). Furthermore, I find it simplistic to argue that most of grassland soil bacteria are dormant if the soil was taken from 0-15 cm depth which is usually very densely rooted. Thus, the soil sampled in this experiment should consist of rhizosphere soil to a large extent where bacteria should be less dormant than in a bulk soil sample. Speaking of resuscitation is a bit far-fetched in this context as well.

L126f: Very interesting result. Please provide the proportion of the active ASVs in comparison to the total abundance per treatment in the text. I am surprised to see in Figure 1 that the total bacterial abundance was as low as 1.0-2.0 E07 which is at least one order of magnitude lower than what we usually observe in grassland soils (E08-E10). How can the authors explain such low bacterial abundances? This also means that in both treatments a large fraction of the total ASVs were also active (>30%). This is a striking finding that should be discussed as it is contrary to other recent studies.

Responses to Comments from the Reviewers

(Manuscript # NCOMMS-23-19532-T)

Reviewer #1 (Remarks to the Author):

A1. *This is an elegant study with noteworthy results, particularly the exploration of microbes involved in the priming effect using qSIP, testing the priming effect in a long-term warming study, and using qSIP data to improve modeling the soil C cycle.*

I believe the work will be of some significance to the field, but my own recommendation would be to develop the manuscript for a soil or modeling journal, as I feel the largest contributions this paper could make are in advancing the thinking within the field rather than trying to cast the results as significant for long-term soil C projections. The main issues I see are

Response: Thank you for your positive comments on our study and your recognition of its significance. In response to the primary concerns you highlighted - "Model Prediction" (A2), "Microbial Data and Priming Effect" (A3), and "Comparative Analysis of Soil C Models" (A4) – we have undertaken thorough revisions to elaborate on them. Please refer to our subsequent detailed explanations for how we have addressed each of these concerns.

A2. *The manuscript overstates the implications of the one-week study of disturbed soils in jars in the lab for soil carbon loss over decades to centuries. For example, the abstract states "warming-enhanced soil priming can lead to considerably more (~52%) soil C loss by the end of this century... soil priming would considerably strengthen the positive feedbacks between the terrestrial C cycle and climate warming." There are many problems with this extrapolation. First, in general, priming is associated with net soil carbon accrual. See <https://www.nature.com/articles/s41467-018-05667-7> for a meta-analysis that shows that the fresh plant inputs are larger than the native C loss, so net soil C accrues. Second, native soil carbon in priming studies is operationally defined as any soil carbon that is not the newly-added carbon that elicits the effect. Yet there is a tendency to assume that this native soil carbon is quite old and thus important for long-term storage. The quoted statement in the abstract implies exactly that, but there is no evidence presented here for disproportionate loss of old carbon. This is true of virtually all priming studies. One limitation of that design element is that extrapolating to modeled long-term soil carbon loss is highly speculative. The soil carbon results from the warming field site study show no indication of the expected decline, but rather a tendency (i.e., not significant) toward a +10% soil C increment caused by 7 years of warming.*

Response: Thank you for pointing this out. Firstly, we would like to clarify a key point: our predictions regarding future carbon loss are based on the MEND model calibrated with field data instead of the 7-day lab incubation data. Specifically, we first integrated the lab incubation datasets into the MEND model, generating a lab-scale MEND (lab-MEND) model to simulate 7-day incubation. This lab-MEND model further played an important role by informing the prior parameter range for a separate field-scale MEND (field-MEND), which assimilated the field warming experiments to simulate soil C decomposition from 2010 to 2016.

About the reviewer's concern on the over extrapolation, firstly, we agree that fresh plant inputs are larger than the native C loss, leading to net soil C accrues during the lab incubation¹. Indeed, our priming data supports the transformation of native C. However, alterations in net soil C of the field are governed by numerous transformative factors, including litter C, root C, and root exudates. These complexities extend beyond the scope of our current data, especially concerning the formulation of new soil C. As a result, although modeling can offer predictions on net soil C changes in the field, the outcomes inherently carry significant uncertainties due to the lack of constraints and intricate underlying mechanisms. Furthermore, given the intrinsic heterogeneity of soil environments and the resulting variations in soil C measurements, identifying and modeling soil C changes within such variations presents challenges^{2,3}. Therefore, we only used the model to predict C loss induced by the priming effect but not the net soil C changes in our original manuscript. To be more rigorous and mitigate the concerns about extrapolation, we have chosen to focus our estimates on the proportion of CO₂ emissions resulting from the priming effect within the broader context of total heterotrophic respiration. Moreover, we have limited our model simulations to the field experimental period (2010–2016). Details are shown as “Model simulations from 2010 to 2016 indicated a notable increase in soil C decomposition under warming, with a 7% rise in priming-induced CO₂ emissions. If our findings can be generalized to other ecosystems over an extended period, soil priming could make considerable contributions to the positive feedbacks between the terrestrial C cycle and climate warming.” (lines 40-44 in Abstract in the revised version), “With the calibrated field-MEND model, we further explored the potential contributions of microbial feedbacks to soil C loss in response to the field experimental warming. Model simulations showed that in situ field warming significantly ($p < 0.001$) increased the 2016 annual average of microbial growth (Extended Data Fig. 15b), the active microbial fraction (Extended Data Fig. 15c), and the decomposition rates of all three soil C pools in the MEND model (Extended Data Fig. 15d). Particularly, the simulated decomposition rate of mineral-associated organic C (MOC, a stable and native C pool with a low reaction rate) increased by 82% under warming (Extended Data Fig. 15d). By tracking the C transformation processes in field, the priming-induced CO₂ emission due to plant fresh C input was projected to contribute 86 % of R_h under control and enhanced by 7 % under warming during 2010-2016. Overall, the model simulation predicted enhanced microbial decomposition and priming processes under warming.” (lines 298-308 in Result in the revised version), and “Using field-MEND, we further obtained the simulated microbial growth rate, microbial active fraction, and the decomposition rates of three SOC pools in 2016. To estimate the priming-induced C loss for the field site, another simulation experiments with the best parameter set were run from 2010 to 2016, assuming that all C input excludes fresh C (i.e., dissolved organic C and particulate organic C degraded by hydrolytic enzymes) for both the control and warmed plots. The priming-induced C loss was defined as $\text{PrimC} = R_h$, normal carbon input - R_h , no fresh carbon. The contribution of priming-induced C loss to R_h was defined as $\text{PrimPercent} = \text{PrimC} / R_h$, normal. The additional contribution of priming-induced C loss to R_h due to warming was calculated as the difference in PrimPercent between the control and warmed plots.” (lines 768-776 in Method part in the revised version).

A3. *The use of ^{13}C -labeled plant material for qSIP allows focus only on those microorganisms that use the ^{13}C labeled material to grow, but not on the organisms that specifically are involved in priming, which as the authors note is "an effect of fresh plant-derived C on native SOC decomposition." But the decomposition of native SOM is where the priming action is. The approach used here does not interrogate priming, per se, the mystery of which is about which organisms increase the oxidation of native SOM. Much of the priming literature invokes the idea that they are not, that priming involves the initial response of "r-selected" taxa (or copiotrophs, depending on the particular argument) that then trigger the response of "K-selected" taxa (or oligotrophs). The approach used here only identifies one group and not the group that is particularly interesting because of their possibly important biogeochemical role. (It is possible to distinguish these groups using qSIP. See <https://doi.org/10.1038/ismej.2017.43>.) For this reason, this current study is not particularly informative about the microbiology of the priming effect.*

Response: We agree with the reviewer that there are two groups of microorganisms involved in consuming the additional SOC (priming effects). The first group comprises microorganisms that not only utilize the ^{13}C labeled straw for their growth but also concurrently consume the additional native SOC. The second group comprises microorganisms that specifically utilize the native SOC without utilizing the added fresh ^{13}C -labeled straw. To address the reviewer's comments, we expanded our statistical analyses beyond the original PLS model presented in our manuscript (**Lines 197-203 and Supplementary text D in the revised version**).

First, we utilized both the Mantel test and Pearson correlation to assess the relationship between the "active to fresh C" microbial community and primed C. Our analysis revealed a significant positive correlation at the family taxonomic level between primed C and six microbial families (e.g., *Planococcaceae*, *Rhizobiaceae*, *Sphingomonadaceae*), as shown in **Supplementary Table 3 in the revised version**. Additionally, at the ASV taxonomic level, two ASVs (ASV_2 and ASV_45) within the active communities displayed a significant correlation with primed C, as detailed in **Supplementary Table 5 in the revised version**. We also observed marginal or significant correlations between primed C and active functional gene groups involved in degrading a wide range of carbon compounds targeted by GeoChip, including starch, hemicellulose, cellulose, chitin, phospholipids, and vanillin/lignin (**Supplementary Table 6 in the revised version**). Our findings align closely with those reported by Morrissey et al.⁴, suggesting that multiple microbial families, stimulated by straw addition, contribute to the soil priming effect. This data further supports our qSIP experimental results' connection to the soil priming effect.

Secondly, we also conducted analyses using the Mantel test and Pearson correlation to examine the correlation between the "inactive to fresh C" community and the primed C. Similarly, our findings revealed a significant positive correlation between primed C and six families within the inactive bacterial communities, including *Acanthopleuribacteraceae*, *Clostridiales_Incertae_Sedis_III*, *Chloroflexaceae*, etc, among others, as listed in **Supplementary Table 4 in the revised version**. At the ASV taxonomic level, we identified four ASVs (ASV_162, ASV_289, ASV_1112, and ASV_1624) from the inactive communities that exhibited a significant correlation with primed C, as shown in **Supplementary Table 5 in the revised version**. These

results suggest that certain taxa within the inactive community, despite not responding to the fresh C addition, are capable of specifically utilizing the native SOC.

Thirdly, we evaluated the average 16S rRNA gene copy numbers in the highly correlated families/ASVs from both the “active to fresh C” and “inactive to fresh C” communities. Our analysis revealed that the average 16S rRNA gene copy number in the active communities was significantly higher compared to the inactive communities (**Supplementary text D: Lines 106-108 and 115-116 in the revised manuscript**). This observation suggests a distinction in ecological strategies between the two groups. Organisms within the active communities, with higher 16S rRNA gene copy numbers, are more likely to be copiotrophic or *r*-strategists. In contrast, organisms in the inactive communities, with lower 16S rRNA gene copy numbers, are more inclined to be oligotrophic or *k*-strategists.

Finally, while we have identified several families or ASVs from the second group (“inactive to fresh C” community) that are highly correlated with the soil priming effect and may potentially specifically utilize native SOC, we acknowledge a limitation in our conclusions due to the absence of ¹⁸O-H₂O qSIP data. Without this data, we cannot definitively confirm that these organisms are actively consuming the additional native SOC. We recognize this as an area for further investigation in our future studies. This limitation is now explicitly noted in **Supplementary text D: lines 139-143** of our revised manuscript, shown as “While we identified several families or ASVs from the “inactive to fresh C” community that are highly correlated with the soil priming effect, the absence of ¹⁸O-H₂O qSIP data means we cannot definitively confirm that these organisms are actively consuming the additional native SOC. This remains an area for further investigation in future studies.”

A4. *As an example of how microbial data can be used to improve soil C models, this work is strong. I'd like to see this more developed, more clearly explained, the sensitivities thoroughly explored, and alternative model formulations formally compared. As it is, while the optimization shows an improved fit using the approach described, there is clearly room to develop this further. Some still argue that classic 3-pool carbon models are "better" in the sense that they are simple and not overly parameterized. Microbial ecological data can rigorously improve this if the data represent actual processes that occur in soil that are represented clearly both in the data and in the model. I can't tell how close this study comes to meeting that high bar, because the model is not sufficiently described nor how the data are treated to compare with model output. These authors could build a stronger paper largely around this point in my view.*

Response: We greatly appreciate your insights and the emphasis on seeing “*the modeling part more developed, more clearly explained, the sensitivities thoroughly explored, and alternative model formulations formally compared*”.

Firstly, to address the development and comparison concern, as suggested, we have undertaken a comparison of our MEND model with the Terrestrial ECOsystem (TECO) model, which divides the soil C pools into 3 classical pools (fast, slow, and passive). We selected the TECO model for a comparative analysis, as it is derived from the classic 3-carbon pool model and its applicability to field sites⁵. Compare to MEND, TECO lacks specific microbial and enzyme pools, which limits its ability to assimilated microbial information. Detailed on TECO model is shown as “TECO is a conventional CENTURY-like model, where the decomposed soil organic C

is proportional to the C pool size (Extended Data Fig. 17). In TECO, Plant biomass enters litter C, which is decomposed into CO₂ and transformed into soil C pools. Soil C pools are then decomposed into CO₂ and transformed into more recalcitrant C. Soil carbon decomposition rates vary with temperature and moisture and are calculated as the product of baseline turnover rate and scaling functions for temperature and water stress. TECO includes the traditional three soil C pools (fast, slow, passive) and can predict ecosystem dynamics. Compared to MEND, TECO does not include specific microbial pools and enzymes, therefore, limits its ability in assimilating microbial trait information⁷²” in Lines 707-715 in Method of the revised version. Specifically, 1), we conducted five model experiments to compare the parameter uncertainty of TECO and different field-MEND models assimilating different information. As a result, TECO can only assimilate Rh data and has the highest parameter uncertainty, while field-MEND assimilating all information reduced the uncertainty by 78% under warming conditions. Detailed results are shown in Lines 263-278 in Result of the revised version, “To determine how microbial genomic information and laboratory data help calibrate ecosystem models, we conducted five model experiments to test different combinations of calibration data for field-MEND and the Terrestrial ECOsystem (TECO) models (Fig. 5d). TECO includes the traditional three soil carbon pools (fast, slow, and passive), but does not explicitly represent microbial pools and their processes⁷². Under both warming and control conditions, TECO had the largest parameter uncertainties on average (Fig. 5d, dark blue bars), potentially due to the lack of microbial and enzyme groups to assimilate additional microbial information. By contrast, the field-MEND, which assimilated both Rh and in situ gene abundance, reduced the parameter uncertainty by 3-67 % (Fig. 5d, yellow bars). The addition of active fraction and lab-MEND derived parameters reduced the parameter uncertainty by 7-44 % and 6-19 %, respectively (Fig. 5d, grey and red bars). The field-MEND, which assimilated all information, had the highest performance and reduced the parameter uncertainty by 32-37% compared with field-MEND assimilating Rh and genes ($p < 0.05$, Fig. 5d, light blue bars). In total, the calibration with all data using field-MEND reduced the uncertainty by 78% compared with TECO under warming condition ($p < 0.01$). These results highlighted that integrating diverse microbial data in ecosystem models can effectively reduce parameter uncertainty.”.

Fig. 5 | d, *The field-MEND and TECO model parameter uncertainty assessed by the Coefficient of Variation (CV). The bars show the mean CV values of the 13 parameters for field-MEND and 10 parameters for TECO. The bars with different colors represent five model experiments and the above table columns of the same color list the information used in calibration process for each model. The + label indicates usage of the information when doing calibration. Lab-MEND derived parameters indicates replacing the*

parameter prior ranges with the parameter uncertainty ranges derived from lab-MEND calibration before calibration of field-MEND, which included five parameters (i.e., p_{EP} , V_g , α ,

K_D , and Y_g). Significance is indicated by *, $0.01 < p \leq 0.05$; **, $0.001 < p \leq 0.01$; and ***, $p \leq 0.001$ with Wilcoxon tests. 2), We further compared prediction accuracy of R_h for TECO and field-MEND, field-MEND showed higher goodness-of-fit ($R^2 = 0.55$ for warming, $R^2 = 0.60$ for control) under warming condition than TECO ($R^2 = 0.45$ for warming, $R^2 = 0.60$ for control, **Extended Data Fig. 14** in the revised version).

Extended Data Fig. 14 Calibration results of lab-MEND, field-MEND and TECO models. **a**, Lab-MEND simulated responses of EPO and EPH vs. observed responses microbial functional gene abundance. **b**, Lab-MEND simulated microbial active fractions vs. observed microbial active fractions. **c**, Field-MEND simulated microbial active fractions vs. observed microbial active fractions. **d**, Field-MEND simulated enzyme concentrations vs. observed in situ microbial functional gene abundance. **e**, TECO simulated in situ R_h vs. observed in situ R_h . EPO, EPH: oxidative enzymes and hydrolytic enzymes for degrading POC in MEND. As gene abundance and enzyme concentrations have different units, they cannot be compared directly. Alternatively, we compared their responses to warming (by dividing the values under warming with values under control) or temporal variation (by scaling it to a standard normal distribution), which would remove the unit differences.

Second, to address the Reviewer's concern on the model is not sufficiently described (clear explanation), we have made the following revisions:

1), We have included a schematic figure (**Extended Data Fig. 18 in the revised manuscript**) that outlines how we integrated both lab- and field-scale data to calibrate the MEND model, The figure is shown as below:

Extended Data Fig. 18 | The flowchart of model calibration and experiments with field-MEND and lab-MEND. Grey boxes mainly represent steps of field-MEND model and green boxes represent steps of lab-MEND. The arrow indicates the results of the former box serve as the input, parameters, or the model for the latter box. The field-MEND and lab-MEND shared similar calibration algorithms (the steps within the red frame) except that field-MEND model use field warming experiment data as well as lab-MEND derived parameters to adjust the initial parameter sets used in calibration. The model experiment helps to determine the best models outside the calibration steps.

2), We have also clarified the definition and objective of lab-MEND and field-MEND at the beginning of the modeling section in the results (**Lines 241-247 in Result of the revised manuscript**), shown as “Here, we initially calibrated the lab-scale MEND (lab-MEND) model using laboratory measurements, including microbial respiration (total CO₂ and ¹³CO₂ fluxes) with or without litter addition, microbial active fraction, and oxidative (EnzCo) and hydrolytic enzyme (EnzCh) concentrations informed by GeoChip-detected functional gene abundances of active communities. Then, we calibrated the field-scale MEND (field-MEND) model using field

measurements, including in situ heterotrophic respiration rates (R_h), monthly and annual gene abundance, and microbial active fraction as well as lab-MEND derived parameters.”.

3), we've included more detailed description of MEND model and explain why it is suitable to use MEND for assimilating microbial data for the qSIP experiments, shown as “MEND describes soil organic C pools, microbial biomass C pools, and enzymes C pools. Plant C first enters three soil organic C pools: two particulate organic carbon (POC) pools and DOC pools. The microbial pool directly takes up C from DOC and produces oxidative (EnzCo) and hydrolytic enzymes (EnzCh) to decompose POC into DOC and mineral-associated organic carbon (MOC). MEND implements the transition of microbes between the two physiological states (active and dormant), which have different C mineralization and enzyme producing abilities. The flux rates are determined by the governing equations (Supplementary Table 8). The allocation of fluxes to different pools are described in Supplementary Table 9. The response functions which scale the flux rates under different pH, temperature and moisture are described in Supplementary Table 10. The definition and initial ranges of parameters used in the equations and functions were described in Supplementary Table 11. MEND was chosen for reproducing C and microbial community patterns in lab and field warming experiments due to its explicit mechanisms explaining priming effects, including stimulated growth of microbial biomass and transition from a dormant to an active state with fresh carbon input.” **In lines 632-646 in Method of the revised version.**

4), We've highlighted the comparison of models with different calibration strategies, which provided selection criteria of data assimilation with different models. Details are shown as “After calibrating the lab-MEND, we incorporated its parameter ranges, microbial genomic information and field warming experiment datasets into a new MEND model called field-MEND (Extended Data Fig. 18). To determine how microbial genomic information and laboratory data help calibrate ecosystem models, we conducted five experiments with various combinations of calibration data for field-MEND and Terrestrial Ecosystem (TECO) models: i). TECO model calibrated with R_h . ii). Field-MEND model calibrated with R_h and microbial genomic information, that is, gene abundances of oxidative (EnzCo) and hydrolytic enzymes (EnzCh). iii). Field-MEND calibrated with R_h , genomic information and microbial active fraction. iv). Field-MEND calibrated with R_h , genomic information and lab-MEND derived parameter uncertainty ranges. v). Field-MEND calibrated with all available information (Fig. 5d).” in **Lines 696-706 in Method of the revised version**, and “We replaced the initial parameter ranges of field-MEND with the calibrated parameter uncertainty from lab-MEND (Supplementary Table 7) before starting the calibration process for the 4th and 5th model experiments. For TECO model, we calibrated 10 parameters, including seven for baseline turnover rates of seven plant biomass, litter and soil C pools as well as three for scaling functions controlling temperature sensitivity and moisture response in C decomposition.” in **Lines 727-731 in Method of the revised version.**

Finally, to address the concern about the exploration of sensitivities, we have made the following revisions:

1), We first clarified the application of parameter sensitivity analysis during previous model development^{6,7}, which identified the key parameters to be calibrated in current study as shown in

lines 724-727 in the revised vision, *“The parameters to be calibrated were selected based on their sensitivities evaluated by Multi-Objective Parameter Sensitivity Analysis (MOPSA) method in previous MEND development^{69,120} (Supplementary Text E).”*

2), Then, we aimed to enhance our model by using different types of real-world data to confirm its accuracy and refine key parameters, going beyond sensitivity analysis. We explored data assimilation strategies and highlighted the sensitivity of parameterization to microbial information (Fig. 5d), an aspect undervalued in ecosystem modeling. Specifically, i) we’ve used different objective functions to capture various features (i.e, magnitude, treatment difference, temporal variation) in data, as shown in **lines 732-744 in the revised version**, *“The model parameters of the field-MEND and TECO were optimized by achieving high goodness-of-fit of model simulations against in situ observations. For each of the observed variables, an objective function was assigned to quantify the model performance between observation and simulation (Supplementary Table 12). The parameter optimization aimed to minimize the overall objective function as described in Eq. 4-5. For this field data assimilation, the coefficient of determination (R^2 , Eq. 8) was used to evaluate the Rh because it is frequently measured, and the absolute values can be directly compared between observations and simulations. The MARE (Eq. 6) was used to evaluate the variables (e.g., microbial biomass, active fraction, and the warming response of enzyme concentration) with only a few measurements, and the absolute values can be directly compared. To capture the temporal variations of enzyme concentrations, the correlation coefficient (r , Eq. 9) was used to evaluate the correlation between observed and simulated values, which could omit the magnitude and unit differences in gene abundance data and simulated enzyme concentration.”*

ii), we’ve assigned weights to data in the sum of objective functions to prioritize their importance in data assimilation, in addition to selecting different objective function metrics, as shown in **Lines 663-676** *“We implemented multi-objective calibration of the model⁸. Each objective evaluates the goodness-of-fit of a specific observed variable, e.g., cumulative CO₂ efflux, or relative gene abundances (Supplementary Table 12). The parameter optimization aims to minimize the overall objective function (J) that is computed as the weighted average of multiple single objectives (Supplementary Table 12)⁹.*

$$J = \frac{\sum_{i=1}^m w_i \cdot J_i}{\sum_{i=1}^m w_i} \quad (\text{Eq.4})$$

$$\sum_{i=1}^m w_i = 1 \text{ with } w_i \in [0,1] \quad (\text{Eq.5})$$

*where m denotes the number of objectives, and w_i is the weighting factor for the i^{th} ($i = 1, 2, \dots, m$) objective (J_i). In the laboratory data assimilation, J_i ($i = 1, 2, 3, 4$) refers to the objective function value for cumulative CO₂ efflux, EnzCo, EnzCh, and active fraction, respectively. Since there are far more cumulative CO₂ efflux observations (e.g., 42 in control and warmed soils) than the other variables and CO₂ efflux is the most important variable in soil C studies; we assign a much higher weighting factor to cumulative CO₂ efflux than the other three objective functions (EnzCo, EnzCh and active fraction), i.e., $w_1 = 5/8$ and $w_2 = w_3 = w_4 = 1/8$.” and **Lines 744-747** *“Similar to the laboratory data assimilation, a much higher weighting factor to Rh ($w_1 = 5/9$) was assigned than the other four objective functions (MBC, active fraction, EnzCo and EnzCh, $w_2 = w_3 = w_4 = w_5 = 1/9$). This approach ensures that the model’s performance in predicting Rh is prioritized, while still considering the other variables.” in the revised version.* These methods enhance the utilization of diverse data types, addressing issues of underrepresentation, overrepresentation, and data noise during calibration.*

iii), We've added more data assimilation results to confirm that our chosen objective function metrics effectively capture various data features in the revised manuscript. The additional results showed how the metrics work on data-model integration, which showed good agreement on lab-MEND (or field-MEND) simulated vs. observed values of gene information and active fractions. Details are shown as “Additionally, the active fractions based on qSIP and the response ratios of the GeoChip-based functional genes in active communities were assimilated into the lab-MEND model (Extended Data Fig. 14), enabling direct constrains on simulated enzyme production and microbial activation processes.” in **Lines 251-254** and “The model simulation achieved a high correlation between the simulated enzyme concentrations and GeoChip-detected oxidative and hydrolytic gene abundances (Pearson correlation $r = 0.59$ for EnzCo; $r = 0.67$ for EnzCh, Extended Data Fig. 14). In addition to the annual GeoChip-detected gene abundances collected from 2010-2016¹⁵, the model also successfully fitted the temporal variations of 22 monthly gene abundance data points for 2012 and 2016 (Extended Data Fig. 16). The model simulation also captured magnitudes and changes in microbial active fractions under control and warming conditions (MARE = 0.12, Extended Data Fig. 14). These results suggested that the field-MEND model fitted the observed microbial dynamics well with strong constrains applied to microbial processes and their controlling parameters.” in **Lines 280-289** in the revised version.

Reviewer #2 (Remarks to the Author):

B1. *The manuscript explored the litter-induced priming effect based on a 7-day incubation experiment with ¹³C labeled straw addition under control and warming treatments. The research theme is important and the scope fits the journal. The method is sound and relevant data analysis and modeling approach are appealing. Though the experiment apparently involves two factors, i.e., temperature and straw addition, the current focus on priming, partly use of the entire data, only discussed the straw-induced respiration change under different temperatures. My first concern lies in the lack of details and clarifications of the experimental design and its associated research objective. Whether this experiment is designated for a priming study or factorial study, justifications are needed simply because your research objective, hypothesis, and experimental design remain essentially and coherently consistent or support each other. This ambiguity is expected for clearance in the revised version. The work showed superior strength in the intention of employing various analyses but their connectivity and crossing support seem poor. Overall, the manuscript is potentially publishable after some major and minor revisions. Line comments were also listed below.*

Response: Thank you for the encouragement and the comment regarding the clarity and coherence of our experimental design and research objectives. The research objectives for this study are: (i) whether and how does experimental warming affect soil priming; (ii) what are the microbial mechanisms underlying soil priming, and (iii) can soil priming and associated microbial mechanisms be incorporated into ecosystem models to improve model performance and reduce model uncertainty? (**Lines 88 -91 in the revised version**). We understand the experimental designing and main text connectivity may not be clear enough due to too much data and information in this study. To address this concern, we have incorporated a schematic figure (**Fig 1a in the revised manuscript**) that visually represents our experimental design and

elucidates the aims of our study. This addition aims to enhance the overall clarity, connectivity, and readability of our paper. We believe that this new figure will effectively address the ambiguities in our original manuscript and better illustrate the interconnectedness of our analyses and research objectives.

Fig. 1 | Study design and objectives. *The schematic of the study design illustrates soil sampling from warmed and control plots, followed by comprehensive analyses of soil and microbial properties and mechanisms, and subsequent field model optimization based on lab-derived data/model. The research objectives of this study are to ascertain: (i) the effects of experimental warming on soil priming; (ii) the microbial mechanisms underlying soil priming; and (iii) the potential for incorporating soil priming and associated microbial mechanisms into ecosystem models to enhance model performance and reduce uncertainty. To address these objectives, eight surface soil samples (0–15 cm depth) were collected in 2016 from warmed (targeted continuous heating at +3°C above ambient temperature) and control plots (n=4) at a long-term (7-year) experimental warming site in a tallgrass prairie ecosystem in the US Great Plains of Central Oklahoma (34°59'N, 97°31'W). Following geochemical measurements and Dissolved Organic Matter (DOM) analysis, the samples were incubated for one week with ¹³C-labeled wild oat (*Avena fatua*) straw powder to simulate plant litter decomposition, with additional treatments involving ¹²C-labeled straw and no straw serving as isotopic control and background, respectively. Active degraders in both warming and control samples were identified using qSIP analysis to further explore the microbial mechanisms underlying soil priming. Subsequently, the lab incubation datasets were integrated into a lab-scale Microbial-ENzyme Decomposition (MEND) model to simulate the 7-day incubation period. This lab-MEND model informed the prior parameter range for a separate field-scale MEND (field-MEND) model, which assimilated field warming experiments conducted from 2010 to 2016 to simulate soil C decomposition. Concurrently, the field-MEND model was compared with the Terrestrial ECOsystem (TECO) model to validate the effectiveness of incorporating microbial data into the MEND model for improving performance and reducing uncertainty.*

B2. Lines 34-37. Too long and too vague. Short sentences are recommended.

Response: We have addressed this concern by revising the sentences to be shorter and more concise, shown as “Here, we showed that long-term warming significantly accelerated soil priming (12.7%) in a temperate grassland. Accordingly, warming altered active bacterial communities and increased their abundances, with 38% of unique active phylotypes detected under warming. The functional genes essential for soil C decomposition were also stimulated, which were strongly linked to priming effects.” in **Lines 34-38 in Abstract of the revised version.**

B3. Lines 41-42. Do you mean model results?

Response: Yes. We have explicitly indicated it in the revised manuscript by changing the sentence to “Model simulations from 2010 to 2016 indicated a notable increase in soil C decomposition under warming, with a 7% rise in priming-induced CO₂ emissions.” in **Lines 40-42** in the revised version.

B4. Line 88. Do you mean microbial-based mechanisms?

Response: Yes, we have explicitly indicated it to be “microbial mechanisms” in **line 89** of the revised manuscript, shown as “(ii) what are the microbial mechanisms underlying soil priming, and”.

B5. Line 94. Do you mean the model already has incorporated soil priming effect and because of the warming-enhanced soil priming effect, different simulation results reached as compared with the model excluding soil priming. The confusion lies in the way how soil priming was included in your model.

Response: Thank you for highlighting this potential source of confusion. Yes, the MEND model we used already incorporates some mechanisms underlying soil priming effects. In this study, we specifically aimed to emphasize that microbial-based mechanisms and information underpinning the warming effects have been incorporated into the MEND model. To improve clarity, we have replaced the phrase “such information” with “microbial-based mechanisms and information” in our revised manuscript (**Line 95 in the revised version**). The revised sentence now reads: “Our results reveal that experimental warming indeed strengthens soil priming via activating various functional populations, and that incorporating microbial mechanisms and information into an ecosystem model significantly reduces model uncertainty and improves the accuracy of predictions (Fig. 1).”(**Lines 93-97 in the revised version**).

B6. Line 374. reference is missing.

Response: Thank you for bringing this to our attention. We have addressed this issue by adding the appropriate references (**Line 404 in the revised version**).

B7. Lines 398-400. *Is this the definition of priming effect? is it the same calculation in control and warming treatments? did you report the respiration from straw and how does it differ between control and warming treatments? It appeared that a report of respiration in different treatments would help given that the experiment is basically a two factor experiment involving temperature and straw addition. Priming as named is just a derived thing from the experiment. The relationship of the entirety experiment and specific focus on priming seems vague in the current form.*

Response: The content from lines 398 to 400 describes our method for calculating the primed C. We applied this calculation consistently across both control and warming treatments. We have indicated this as “For all samples, microbial respiration was calculated as the sum of the amount of $^{12}\text{CO}_2$ and $^{13}\text{CO}_2$.” in **line 435 in the revised version**.

Regarding your query on straw respiration, we did not specifically delineate the respiration derived solely from the straw in our initial manuscript. To clarify:

In the control samples:

The basal soil respiration was at $22.2 \pm 2.1 \mu\text{g C/g soil}$;

The litter-derived CO_2 was $182.7 \pm 16.4 \mu\text{g C/g soil}$.

In the warmed samples:

The basal soil respiration was at $24.8 \pm 4.0 \mu\text{g C/g soil}$, comparable to that of the control samples ($p > 0.050$, permutation ANOVA).

The litter-derived CO_2 was at $208.0 \pm 10.4 \mu\text{g C/g soil}$, significantly higher than that of the control samples ($p < 0.01$, permutation ANOVA).

To address this gap and better elucidate the relationship between the entirety of our experiment and our specific focus on priming, we have incorporated the above detailed information in **Supplementary Text A (Lines 8-18)** as “By analyzing $^{13}\text{CO}_2$, we can differentiate CO_2 derived from the added litter from that derived from native soil organic C. In the control samples, the 7-day cumulative litter-derived CO_2 amounted to $182.7 \pm 16.4 \mu\text{g C/g soil}$, and the native soil-derived CO_2 (i.e., native soil respiration) reached $143.3 \pm 8.9 \mu\text{g C/g soil}$, substantially exceeding basal soil respiration (i.e., soil respiration without litter addition, $22.2 \pm 2.1 \mu\text{g C/g soil}$) (Fig. 2b & Extended Data Fig. 2). In the warmed samples, basal soil respiration ($24.8 \pm 4.0 \mu\text{g C/g soil}$) was similar to that of the control samples, while the litter-derived CO_2 rose to $208.0 \pm 10.4 \mu\text{g C/g soil}$ ($p < 0.01$, permutation ANOVA) and native soil respiration rose to $160.4 \pm 9.3 \mu\text{g C/g soil}$ ($p < 0.05$, permutation ANOVA). Consequently, microbial respiration in the warmed samples was significantly higher ($p < 0.05$, permutation ANOVA) than in the control samples, with an increase of $14.2\% \pm 12.8\%$ (Fig. 2b & Extended Data Fig. 2).”.

”. We believe that the additions to supplementary text A, in conjunction with Figure 1a, enhance the coherence and connection between our overarching experimental design and the nuanced focus on priming.

Reviewer #3 (Remarks to the Author):

C1. Disclaimer: *I was asked to specifically comment on the soil microbiota characterisation using high-throughput sequencing in this study. While reviewing I have also identified other points that I have listed below.*

Response: Thank you for your expertise on soil microbiota characterization using high-throughput sequencing. We appreciate your thorough review and the additional points you have raised. We have made efforts to address each of your comments as detailed below.

Methods:

C2. L323: *Port-Pulaski-Keokuk complex is not a soil type or at least none in a commonly used classification system. Please provide the soil type according to USDA or FAO/WRB.*

Response: Thank you for the comment. Upon further verification using the USDA-NCSS soil survey data from UC Davis (<https://casoilresource.lawr.ucdavis.edu/>), our warming field's soil classification is identified as *Mollisols* (Suborder: *Ustolls*). We have updated this information as “The soil classification of the site is *Mollisols* (Suborder: *Ustolls*), which is a well-drained soil formed in loamy sediment on flood plains” in **line 352** of our revised manuscript.

C3. L326: *I am surprised by this statement as a coarse soil like the one presented here (close to sandy loam in the texture triangle) is usually considered to have a medium to low water holding capacity. Also please provide the value for pH here. Later on the authors describe the measurement of pH so I wonder why this information was provided twice?*

Response: Thank you for pointing this out. We agree that our soil, with a 37% water holding capacity (WHC), is typically categorized as having medium to low water holding capacity, especially when compared with soils like clay or peaty types. Our intention in using the term “high” was to emphasize that a 37% WHC is relatively elevated in comparison to standard loamy soils, which commonly range between 10% to 20%. We recognize that the phrasing may have caused confusion, and to ensure clarity, we have opted to remove the term “high” from the text, shown as “The soil has an available water holding capacity of 37%” in **line 355** in the revised manuscript.

Meanwhile, thanks for your keen observation regarding the presentation of pH values in our manuscript. As suggested, we have added and emphasized the pH value from 2009 as “The soil has an available water holding capacity of 37%, neutral pH (6.64 ± 0.24 in 2009), and a deep (about 70 cm), moderately penetrable root zone” in **line 355** in the revised manuscript. This initial pH value, as referenced in our original manuscript on line 323 (ref 53), pertains to measurements from when our warming site was initiated in 2009. In contrast, the subsequent pH measurement corresponds to soil samples taken in 2016. Our objective was to assess any changes in pH after seven years of warming, given the potential significant role pH plays in soil microbial processes. To further remove the confusion here, we have emphasized the first pH value is determined in 2009 on **line 356** in our revised version.

C4. L327: please provide more information to the statement of limitation. How low are the C/N values and who is actually limited by C and N (I guess soil microbes?). I am not sure soil itself can be viewed as limited by C and N.

Response: Thanks for bringing this to our attention. Yes, we agree that “soil itself can not be viewed as limited by C and N”. Our intent was to convey that the soil microbes could be limited by N. Recognizing the imprecision in our original statement "The soil in this site is both C and N limited," we have decided to remove it from the revised version. Furthermore, to provide a clearer context, we've included the C/N value of the oat straw as "The addition of straw powder to soil with a high C/N ratio (C/N ratio: 24)" on **line 357** of our revised manuscript.

C5. L360: I am not an expert on priming but does priming induced by milled and added litter represent priming after all in terms of magnitude? If not, how can the authors be convinced that their upscaling is useful?

Response: Thank you for your insightful questions. Please allow us to elucidate on the methodological approach and the rationale behind the priming effect and then answer your questions:

To begin with, it's worth noting that there is no universally agreed-upon criteria for either the quality (simple vs. complex) or quantity of carbon introduced when studying the priming effects derived from lab incubation experiments¹⁰. Previous study¹¹ found that the “priming effect” is a saturating function of fresh carbon inputs, being only marginally influenced by initial N availability when using straw as the carbon, and another study¹² indicated that under the same quantity of fresh C, the variability in the priming effect is least in grasslands compared to other ecosystems.

With these considerations in mind, we made the following choices for our priming effect study:

1. **Quality of Carbon:** Both ¹³C-glucose (simple) and ¹³C-straw (complex) are conventionally used carbon substrates in lab-based priming effect experiments¹⁰. Our specific choice of ¹³C-straw was driven by the study's objective: to examine the litter-derived priming effect in response to warming. This complex carbon source is more representative of natural litterfall, as highlighted in our original paper.

2. **Quantity of Carbon:** We incorporated 0.1 g of straw with 5g of soil, resulting in an equivalent C amount of 10 mg C /g dry soil. The choice of this C amount is based on:

- By reviewing previous research which utilized complex carbon (e.g., straws or leaves) to study priming effects, the typical range for C addition amounts is between 2.5 and 12 mg C/g soil^{11,13-18}.

- SIP demands a short-term incubation to avoid cross-feeding. The straw is degraded and assimilated much slower by bacteria compared to glucose, as the turnover time for straw is ~10⁴ to 10⁶ longer than the glucose^{19,20}. A previous study using the qSIP to study the priming used 1.25 mg glucose (equivalent to 0.5 mg C/g soil) as the fresh carbon for 1 g soil incubation⁴. Given these, a relatively higher amount of ¹³C-straw is essential for ensuring efficient ¹³C

incorporation into the DNA of active bacteria and producing a discernible “¹³C-incorporation signal” for identification of active bacteria during 7-day incubation.

In response to your queries, we believe there is no lab-incubation derived priming study that can fully emulate the intricacies of natural priming, irrespective of the carbon source or amount used. However, lab-derived insights into the priming effect and its underlying microbial mechanisms can be useful for ecosystem model development and parameterization. As we noted in our original manuscript, inclusion of microbial data offers a more refined constraint on vital microbial parameters influencing SOC (lines 255-258 in original manuscript). This improved calibration, further integrating with the field data, can significantly curtail model uncertainty.

Finally, recognizing this one-week incubation may not capture the effects of more recalcitrant carbon sources, and pondering whether smaller straw additions would yield comparable results for the priming effect in response to warming, we established an additional 63-day incubation experiment with less amount of C addition (equivalent to 3 mg C /g soil), shown as *“The SIP experiment requires a short-term incubation to minimize cross-feeding. Accordingly, we set the incubation period at one week. However, this one-week incubation may not capture the effects of more recalcitrant carbon sources, such as lignin, on priming. Moreover, while previous studies have used similar or even higher amounts of complex C to assess priming effects, it remains unclear whether smaller C additions would yield comparable results, especially for the priming effect in response to warming. Therefore, to account for the possibility of continued carbon processing beyond the initial seven-day incubation period, we established an additional 63-day incubation experiment. In this extended experiment, we aimed to assess the priming effect of both warming and control samples with reduced straw addition 0.33 g of ¹³C-straw in 5 g of soil (equivalent to 3 mg C /g dry soil) over a longer timeframe. As expected, the general patterns of the priming effects were consistent between the 63-day incubation and the 7-day incubation experiments (Fig. 2b, Extended Data Fig. 2 & Fig. 3), but the magnitudes of positive priming effects were different in the two experiments due to the changes in quantity of oat straw during the experimental periods. The overall priming effect in the 63-day incubation experiment was significantly higher ($p < 0.050$, permutation ANOVA) for the soil samples under warming than control, with an increase of $27.8\% \pm 8.1\%$ (Extended Data Fig. 3).”* **in Supplementary text A (Lines 19-34) in the revised version.** Results confirmed that the priming effect patterns observed were consistent between the 63-day and the 7-day incubations.

In light of these considerations, we have added more details in the method part, shown as *“Given that cellulose and hemicellulose constitute over 65% of oat straw’s composition⁶² and hydrolyze faster than lignin, prolonged incubation could lead to severe cross-feeding issues among microbial community members^{90,91}. Thus, to minimize the potential for cross-feeding and enhance the accuracy of subsequent quantitative stable isotope probing (qSIP) experiments^{60,90-92}, the incubation period was set to seven days in this study. This strategy ensures a reliable identification of carbon decomposers. Based on reviewing previous studies which utilized complex carbon (e.g., straws or leaves) to study priming effects^{32,39,93-97}, the typical range for C addition amounts is between 2.5 mg C/g soil and 12 mg C/g soil. Additionally, the straw is degraded and assimilated much slower by bacteria compared to glucose, as the turnover time for straw is $\sim 10^4$ to 10^6 longer than the glucose^{98,99}. A previous study using the qSIP to study the priming used 1.25 mg glucose (equivalent to 0.5 mg C/g soil) as the fresh carbon for 1 g soil*

incubation⁶³. Given these, a relatively higher amount of ¹³C-straw is essential for ensuring efficient ¹³C incorporation into the DNA of active bacteria and producing a discernible "¹³C-incorporation signal" for identification of active bacteria during 7-day incubation. Therefore, we decided to add 0.1 g of straw to 5g of soil (10 mg C/g dry soil) for both determining the priming effect and efficiently labeling DNA of bacteria for qSIP.

However, one week incubation might miss the information of whether more recalcitrant C such as lignin can affect priming. Concurrently, whether smaller straw additions would yield comparable results for the priming effect in response to warming? Thus, considering the potential for continued carbon processing beyond a seven-day incubation period, we also established another 63-day incubation experiment. This incubation experiment was set up exactly as above described, with the only differences being the addition of C at 3 mg C/g soil and an extended incubation period of 63 days.” In **Lines 402-424 in Method of the revised version**) We hope this offers a clearer picture of our methodology and its relevance in the broader context of ecosystem modeling.

C6. L364: *The amount of straw added represents 2% C per g soil (unclear if fresh or dry weight). What was the original C content and how much C (in % to soil dry weight) did the soil used for incubations contain?*

Response: The C and N contents of the straw are 40.10% ± 0.08% and 1.70% ± 0.05%, respectively. This information has been added as “The C and N contents of the straw are 40.10% ± 0.08% and 1.70% ± 0.05%, respectively, determined by the Soil, Water, and Forage Analytical Laboratory at Oklahoma State University, OK, USA.” in **Lines 391-393** of our revised version.

We incorporated 0.1 g of straw with 5g of soil, resulting in an equivalent C amount of 10 mg C /g dry soil. This information has been added as “(i) with 0.1 g of ¹³C-straw in 5 g of soil (equivalent to 10 mg C/g dry soil) as isotopic treatment, (ii) with 0.1 g of ¹²C-straw in 5 g of soil (equivalent to 10 mg C/g dry soil) as isotopic control” in **Lines 396-397** of our revised version.

The choice of this C amount is based on, which we mentioned in the response to comment C5:

- By reviewing previous research which utilized complex carbon (e.g., straws or leaves) to study priming effects, the typical range for C addition amounts is between 2.5 mg C/g soil and 12 mg C/g soil^{11,13-18}.

- SIP demands a short-term incubation to avoid cross-feeding. The straw is degraded and assimilated much slower by bacteria compared to glucose, as the turnover time for straw is ~E04 to E06 longer than the glucose^{19,20}. A previous study using the qSIP to study the priming used 1.25 mg glucose (equivalent to 0.5 mg C/g soil) as the fresh carbon for 1 g soil incubation⁴. Given these, a relatively higher amount of ¹³C-straw is essential for ensuring efficient ¹³C incorporation into the DNA of active bacteria and producing a discernible "¹³C-incorporation signal" for identification of active bacteria during 7-day incubation.

Therefore, we added 0.1 g of straw to 5g of soil (10 mg C /g dry soil) for both determining the priming effect and efficiently labeling DNA of bacteria for qSIP. To address this concern, we

have added above information in **Lines 402-424** in our revised version, which has been shown in above (Response to C5).

C7. L374: *Reference missing as indicated by '(ref)'. Also I don't understand the argument here. The authors argue that fermentation products from switchgrass can be produced by pure cultures in 5 days. The authors want to prevent cross-feeding and keep the incubation for 7 days. Does that mean that the authors assume the decomposition takes longer in the soil setting as compared to the pure culture study? I believe that 7 days can still be considered to be fine but the argument above was not clear to me.*

Response: Thank you for highlighting this point. Our intention was not to suggest that decomposition in a soil setting takes longer than in a pure culture study. Rather, we intended to illustrate that if a single bacterial strain can initiate the degradation of straw within 5 days, then a 7-day incubation period should be sufficient for microbial communities in the soil to begin this process. We acknowledge that our original argument may have been unclear. To remedy this, we've removed that particular argument from our revised version and have also inserted the missing references where indicated, shown as “prolonged incubation could lead to severe cross-feeding issues among microbial community members^{89,90}” on **Line 404 in the revised version**.

C8. L381: *Why are these results presented in the Methods section?*

Response: Thank you for pointing this out. We have now relocated this information to **Supplementary Text A (lines 19-34)** in revised version.

C9. L412: *I am surprised to see that the DNA concentrations were within a range of 2.6 ng/μL (stddev or err?) across all samples. I wish I would have ever been able to extract DNA from soil in such a reproducible manner.*

Response: Thank you for highlighting this point. The value given is indeed the standard error (s.e.). With our set of 24 soil samples, which includes eight C-13 samples (four for warming and four for control), eight C-12 samples, and eight background samples, the standard deviation works out to be 12.7 ng/μL. To clarify and prevent any confusion, we have changed the s.e. to s.d., and explicitly mentioned that this value represents the s.d., shown as “DNA concentrations were 49.1 ± 12.7 ng/μl (mean ± s.d., n =24)” on **line 449** of the revised version.

C10. L431: *Please provide references for the primers used (Caporaso et al. 2011?).*

Response: Thank you for pointing it out. We have added the reference as suggested on **line 469** in our revised version.

C11. L464: *it does not become clear to me what the “random resampling” approach means and unfortunately the reference provided here does not give more detailed information (it is pretty much the same sentence there). Did you multiply the relative abundance of each ASV with the gene copies per sample? Please clarify and modify in the manuscript.*

Response: Thank you for bringing this to our attention. To clarify, we did not multiply the relative abundance of each ASV with the gene copies per sample. Instead, given we used the same amount of DNA for each sample, the ASV table was normalized by rarefying the sequence counts of each fraction based on the total 16S rRNA gene copies in corresponding fraction determined by qPCR. The “rarefy” function in the R package “vegan” was employed for the rarefaction. As a result, a normalized ASV table encompassing all fractions was produced. Based on your feedback, the details regarding the generation of this ASV table have been included in **lines 500-504** of the revised manuscript, shown as “Given we used the same amount of DNA for each sample, the ASV table was normalized by rarefying the sequence counts of each fraction based on the total 16S rRNA gene copies in corresponding fraction determined by qPCR. The “rarefy” function in the R package “vegan” was employed for the rarefaction¹⁰⁴.”.

C12. L525: *where does this data come from? It seems it is not described in the presented manuscript as amplicon sequencing is only described for the qSIP fractions. How would the authors infer activity of bacteria from standard MiSeq data?*

Response: Thank you for pointing it out. The 16S rRNA gene sequencing data is obtained from our previous paper (ref 53 in our original manuscript), which is available in the NCBI Sequence Read Archive under project no. PRJNA331185. Since we have identified which ASVs are the potential active degraders, we can use the unique representative sequence from those active ASVs to search for the same ASVs from the 16S rRNA gene sequencing data spanning 2010-2016. To address this concern, we have indicated that “Since we have identified which ASVs are the potential active degraders in warming or control groups, the unique representative sequences from those “active” ASVs were used to search for the same ASVs from the 16S rRNA gene sequencing data spanning 2010-2016 (The data is available in the NCBI Sequence Read Archive under project no. PRJNA331185)¹⁵. The linear mixed-effects model (LMM) in the “lme4” package was used to determine warming effects on the relative abundance of those “active ASVs” during 2010–2016¹⁵” in **lines 576-580** in our revised version.

Additional comments:

C13. L122-126: *The reference provided is not complete in the bibliography. Also I find this statement a bit misleading as it leaves out fungi completely which are commonly known to be very important decomposers of soil litter and they should be rather abundant in grassland soils as well (AMF, saprotrophs, etc.). Furthermore, I find it simplistic to argue that most of grassland soil bacteria are dormant if the soil was taken from 0-15 cm depth which is usually very densely rooted. Thus, the soil sampled in this experiment should consist of rhizosphere soil to a large extent where bacteria should be less dormant than in a bulk soil sample. Speaking of resuscitation is a bit far-fetched in this context as well.*

Response: Thank you for highlighting this. Based on the phospholipid fatty acid (PLFA) analysis²¹ of our field soil samples, the biomass ratio of fungi to bacteria is 0.055 ± 0.036 for warming plots and 0.043 ± 0.025 for control plots. This data suggests that bacterial biomass is much higher than the fungi in our warming field. Nevertheless, we recognize the limitations of the PLFA method, which might lead to potential underestimations of fungal biomass. Furthermore, even with lower biomass relative to bacteria, fungi undeniably play a pivotal role in

litter decomposition in soils. We concur with the reviewer's observation that the original sentence may be misleading. To rectify this, first, we have rephrased the first sentence by citing our current PLFA result (**Lines 124-127, revised version**), shown as "Based on the phospholipid fatty acid (PLFA) analysis, the biomass ratio of fungi to bacteria is 0.055 ± 0.036 for warming plots and 0.043 ± 0.025 for control plots⁵⁶, suggesting that bacterial biomass is much higher than fungi in our field site.". Second, we have revised the text to emphasize the significant role of fungi in litter decomposition, shown as "Finally, despite the biomass ratio suggesting higher bacterial biomass, fungi undeniably play a pivotal role in litter decomposition." **In Supplementary text C: Lines 84-86 of the revised version.** Additionally, we've highlighted the need for future investigations to identify "active" fungi in the decomposition process, shown as "While our study focused primarily on bacterial contributions, it is worth noting that a holistic understanding of soil processes would benefit from a balanced exploration of both bacterial and fungal roles." in **Supplementary text C: Lines 86-88 of the revised version.**

In response to the second question, the definition and categorization of microbial activity states are essential to our discussion. Blagodatskaya and Kuzyakov's review²² delineates the microbial biomass in soil into three primary states: active, dormant, and dead. In their definition, the dormant state encompasses various resting forms that exhibit significantly reduced respiration and endogenous metabolism over prolonged periods, such as spores. On the other hand, the active state encompasses microbial biomass that:

- 1) is growing and reproducing;
- 2) readily responds to substrate input (e.g., by respiration, producing enzymes).

Consequently, the active state has two sub-states: truly active microorganisms, which are growing (representing 0.1%-5% of total microbial biomass), and potentially active microorganisms, which can readily respond to substrate input (comprising 10 to 60% of total microbial biomass) and immediately change to truly active state after substrate input²².

Therefore, our use of the term "dormant" on line 124 of our original manuscript encapsulates both genuinely dormant and potentially active microorganisms. To clarify this aspect, following with the Reviewer's suggestion, we have replaced the word "resuscitation" with "responses", deleted the confusion word "dormant", and rephrased the sentence to "Consequently, in this study, we focused on examining the responses of active bacterial communities to climate warming. To this end, we extracted DNA after a 7-day incubation and employed quantitative stable isotope probing (qSIP) to identify and analyze the active bacterial community (Extended Data Fig. 4)⁵⁷" in **Lines 127-130** of the revised version.

C14. L126f: *Very interesting result. Please provide the proportion of the active ASVs in comparison to the total abundance per treatment in the text. I am surprised to see in Figure 1 that the total bacterial abundance was as low as 1.0-2.0 E07 which is at least one order of magnitude lower than what we usually observe in grassland soils (E08-E10). How can the authors explain such low bacterial abundances? This also means that in both treatments a large*

fraction of the total ASVs were also active (>30%). This is a striking finding that should be discussed as it is contrary to other recent studies.

Response: Thank you for your insightful observations. We apologize for the oversight in presenting the bacterial abundance in Figure 1c. The original abundance presented in Figure 1c actually was the sum of the 16S rRNA gene abundance of all fractions in each sample, which was derived from the qSIP analysis²³. As a result, the unit should not be “16s rRNA gene copies / g soil” but rather “16s rRNA gene copies/μl resuspension solution”. To obtain the total number of 16S rRNA gene copies per gram of soil for each sample (copies/g soil), the sum of the 16S rRNA gene abundance of all fractions (copies/μl) should be scaled by constant factors such as the resuspension volume, DNA elution volume, the amount of soil used for DNA extraction, and the volume of DNA used for Density-gradient ultracentrifugation, as described by the following equation. After making this adjustment, the bacterial abundance (copies/g soil) for all samples aligns with the E08 range (Warming: $8.22 \times 10^8 \pm 1.33 \times 10^8$; Control: $5.77 \times 10^8 \pm 9.27 \times 10^7$), which matches the typical observations in grassland soils.

To clarify this point, we have updated Fig. 2c (revised version) and incorporated details and the equation, shown as “To determine the total number of 16S rRNA gene copies per gram of soil for each sample (copies/g soil), the total number of 16S rRNA gene copies in each fraction (copies/μl) was scaled by factors such as the resuspension volume, DNA elution volume, the amount of soil used for DNA extraction, and the volume of DNA used for Density-gradient ultracentrifugation, as described by the following equation (Eq. 2). The relative abundance of each ASV was then calculated by dividing its sequence reads by total sequence reads from above normalized ASV table. From this, the “absolute” abundance of each ASV in each sample was derived by multiplying the total number of 16S rRNA gene copies by its corresponding relative abundance. This “absolute abundance” could further be adjusted based on the copy number of the 16S rRNA gene per cell for each ASV. The active bacterial abundance was then computed by summing the “absolute” abundances of ASVs identified as active carbon decomposers.

$$A_{total} = \frac{\sum_{i=1}^n (a_i) \times V_e \times V_r}{V_d \times M}$$

$A_{total,soil}$ Total copy numbers in soil sample (copies/g soil).

i Each density fraction

n Number of fractions

a 16S rRNA gene abundance in each density fraction (copies / μl)

V_d Volume of DNA used for Density-gradient ultracentrifugation (μl)

V_e Elution volume of DNA extraction (μl)

V_r Resuspension volume of DNA precipitation (μl)

M The amount of soil used for DNA extraction (g) ” in **Lines 516-536** in Methods section in the revised manuscript.

In response to the active fraction issue, we appreciate the reviewer's interest in the proportion of active ASVs. As suggested, we have provided the proportion of active ASVs relative to total abundance for both the warming ($65\% \pm 3\%$) and control ($53\% \pm 8\%$) groups, shown as “The proportion of active ASVs relative to the total abundance is $65\% \pm 3\%$ for the warming group

and 53% ± 8% for the control group.” in our **Supplementary Text C: Lines 50-51**. It's pertinent to note that our observations regarding active fractions (>30%) align with extant research²² rather than presenting any stark departure from it. As detailed in **Comment C.13**, the appropriate definition and categorization of microbial activity states remain pivotal to our discussion (Please see our responses in C.13 for details). In short, the active group has two sub-groups²²:

1. **Truly Active Microorganisms:** These are organisms that grow independent of readily available substrates, and they typically encompass a modest 0.1%-5% of the entire microbial biomass in soil environments without these substrates;
2. **Potentially Active Microorganisms:** These organisms are predisposed to respond to substrate inputs, swiftly transitioning to a genuinely active state upon exposure. Notably, they represent a substantial chunk, ranging from 10% to 60% of the overall microbial biomass.

Upon introducing fresh carbon, not only the inherently active microbes but also a significant subset of the potentially active ones, along with some from the dormant category, respond to the substrate input, adopting a truly active role. This places our active fraction observation, around ~60%, in harmony with established literature^{22,24}.

Moreover, it's noteworthy that quantitatively distinguishing between active and dormant biomass, and identifying the microorganisms that are ecologically consequential in their active contributions to ecosystem functionalities, is a sophisticated challenge^{22,25}. A myriad of techniques exists to ascertain active microorganisms in soil^{22,26-30}, including but not limited to:

- Plate count and microbial cultures
- Direct microscopy paired with cell staining
- DNA and RNA sequence comparisons
- Stable isotope probing and specific ¹³C analysis
- Techniques rooted in respiration and substrate utilization

The proportion of activity deduced from each method indeed varies. The average active fraction typically converges around ~50%, as elucidated in Figure 5 of Blagodatskaya and Kuzyakov's review²². A recent study by Papp et al. using the qSIP with H₂O¹⁸ technique illuminated that, on average, a whopping 94% of soil taxa manifested metabolic activity³¹. Also, using the ratio between basal and substrate induced respiration analysis, Blagodatskaya and Kuzyakov found, in soils enriched with plant residues, manures, or in the rhizosphere, most of the microbial community is active rather than dormant²². Hence, our determination of the active fraction via qSIP-¹³C-straw sits comfortably within the documented range.

We hope this provides a clearer perspective on our approach and findings.

References:

- 1 Liang, J. *et al.* More replenishment than priming loss of soil organic carbon with additional carbon input. *Nature communications* **9**, 3175 (2018).
- 2 Sulman, B. N. *et al.* Multiple models and experiments underscore large uncertainty in soil carbon dynamics. *Biogeochemistry* **141**, 109-123 (2018).
- 3 Hungate, B. A., Jackson, R. B., Field, C. B. & Chapin, F. S. Detecting changes in soil carbon in CO₂ enrichment experiments. *Plant and Soil* **187**, 135-145 (1995).
- 4 Morrissey, E. M. *et al.* Bacterial carbon use plasticity, phylogenetic diversity and the priming of soil organic matter. *The ISME journal* **11**, 1890-1899 (2017).
- 5 Wieder, W. R. *et al.* Explicitly representing soil microbial processes in Earth system models. *Global Biogeochemical Cycles* **29**, 1782-1800 (2015).
- 6 Wang, G., Post, W. M. & Mayes, M. A. Development of microbial-enzyme-mediated decomposition model parameters through steady-state and dynamic analyses. *Ecological Applications* **23**, 255-272 (2013).
- 7 Jian, S. *et al.* Multi-year incubation experiments boost confidence in model projections of long-term soil carbon dynamics. *Nature communications* **11**, 1-9 (2020).
- 8 Wang, G. S., Huang, W. J., Zhou, G. Y., Mayes, M. A. & Zhou, J. Z. Modeling the processes of soil moisture in regulating microbial and carbon-nitrogen cycling. *J Hydrol* **585** (2020).
<https://doi.org/ARTN124777>
- 10.1016/j.jhydrol.2020.124777
- 9 Wang, G. S. & Chen, S. L. A review on parameterization and uncertainty in modeling greenhouse gas emissions from soil. *Geoderma* **170**, 206-216 (2012).
<https://doi.org/10.1016/j.geoderma.2011.11.009>
- 10 Blagodatskaya, E. & Kuzyakov, Y. Mechanisms of real and apparent priming effects and their dependence on soil microbial biomass and community structure: critical review. *Biology and Fertility of Soils* **45**, 115-131 (2008).
- 11 Guenet, B., Neill, C., Bardoux, G. & Abbadie, L. Is there a linear relationship between priming effect intensity and the amount of organic matter input? *Applied Soil Ecology* **46**, 436-442 (2010).
- 12 Luo, Z., Wang, E. & Sun, O. J. A meta-analysis of the temporal dynamics of priming soil carbon decomposition by fresh carbon inputs across ecosystems. *Soil Biology and Biochemistry* **101**, 96-103 (2016).
- 13 Wu, L. *et al.* Soil organic matter priming and carbon balance after straw addition is regulated by long-term fertilization. *Soil Biology and Biochemistry* **135**, 383-391 (2019).
- 14 Ye, R., Doane, T. A., Morris, J. & Horwath, W. R. The effect of rice straw on the priming of soil organic matter and methane production in peat soils. *Soil Biology and Biochemistry* **81**, 98-107 (2015).
- 15 Conrad, R., Klose, M., Yuan, Q., Lu, Y. & Chidthaisong, A. Stable carbon isotope fractionation, carbon flux partitioning and priming effects in anoxic soils during methanogenic degradation of straw and soil organic matter. *Soil Biology and Biochemistry* **49**, 193-199 (2012).
- 16 Zhu, L.-x., Xiao, Q., Shen, Y.-f. & Li, S.-q. Effects of biochar and maize straw on the short-term carbon and nitrogen dynamics in a cultivated silty loam in China. *Environmental Science and Pollution Research* **24**, 1019-1029 (2017).
- 17 Chen, X. *et al.* Resistant soil carbon is more vulnerable to priming effect than active soil carbon. *Soil Biology and Biochemistry* **168**, 108619 (2022).
- 18 Zhang, X., Han, X., Yu, W., Wang, P. & Cheng, W. Priming effects on labile and stable soil organic carbon decomposition: Pulse dynamics over two years. *PloS one* **12**, e0184978 (2017).
- 19 Hill, P. W., Farrar, J. F. & Jones, D. L. Decoupling of microbial glucose uptake and mineralization in soil. *Soil Biology and Biochemistry* **40**, 616-624 (2008).
- 20 Pal, D. & Broadbent, F. Kinetics of rice straw decomposition in soils. Report No. 0047-2425, (Wiley Online Library, 1975).

- 21 Wu, L. *et al.* Reduction of microbial diversity in grassland soil is driven by long-term climate warming. *Nature microbiology* **7**, 1054-1062 (2022).
- 22 Blagodatskaya, E. & Kuzyakov, Y. Active microorganisms in soil: critical review of estimation criteria and approaches. *Soil Biology and Biochemistry* **67**, 192-211 (2013).
- 23 Hungate, B. A. *et al.* Quantitative microbial ecology through stable isotope probing. *Appl. Environ. Microbiol.* **81**, 7570-7581 (2015).
- 24 Hund, K. & Schenk, B. The microbial respiration quotient as indicator for bioremediation processes. *Chemosphere* **28**, 477-490 (1994).
- 25 Ellis, R. J., Morgan, P., Weightman, A. J. & Fry, J. C. Cultivation-dependent and-independent approaches for determining bacterial diversity in heavy-metal-contaminated soil. *Applied and environmental microbiology* **69**, 3223-3230 (2003).
- 26 Bölter, M., Bloem, J., Meiners, K. & Möller, R. Enumeration and biovolume determination of microbial cells—a methodological review and recommendations for applications in ecological research. *Biology and Fertility of Soils* **36**, 249-259 (2002).
- 27 Joergensen, R. G. & Wichern, F. Quantitative assessment of the fungal contribution to microbial tissue in soil. *Soil Biology and Biochemistry* **40**, 2977-2991 (2008).
- 28 Musat, N., Foster, R., Vagner, T., Adam, B. & Kuypers, M. M. Detecting metabolic activities in single cells, with emphasis on nanoSIMS. *FEMS microbiology reviews* **36**, 486-511 (2012).
- 29 Breeuwer, P. & Abee, T. Assessment of viability of microorganisms employing fluorescence techniques. *International journal of food microbiology* **55**, 193-200 (2000).
- 30 Hartmann, A. *et al.* Microbial community analysis in the rhizosphere by in situ and ex situ application of molecular probing, biomarker and cultivation techniques. *Plant surface microbiology*, 449-469 (2004).
- 31 Papp, K. *et al.* Quantitative stable isotope probing with H₂¹⁸O reveals that most bacterial taxa in soil synthesize new ribosomal RNA. *The ISME journal* **12**, 3043-3045 (2018).

Reviewers' Comments:

Reviewer #1:

Remarks to the Author:

I appreciate the authors' efforts in revising the manuscript and in the additional modeling. This part of the paper is improved. I wish there could be a more rigorous sensitivity analysis and model comparison, to probe more deeply the role of priming in processes involved in net C storage, and to explore the implications of the findings for long-term C cycling. Perhaps here, can the authors clarify whether the reduced parameter uncertainty could be a result of more parameters (and thus the potential for an over-fitted model)?

I'm concerned readers will be confused about what these findings say about the effect of priming on net soil C accumulation or loss. Are the authors arguing that warming-induced priming is a net source of C loss? Does that statement account for the contributions of microbial growth and death during priming to C storage? In the response letter, the authors note that "we only used the model to predict C loss induced by the priming effect but not the net soil C changes in our original manuscript." But the conclusions still lean on the priming as a part of the climate-carbon positive feedback:

"Model simulations from 2010 to 2016 indicated a notable increase in soil C decomposition under warming, with a 7% rise in priming-induced CO₂ emissions. If our findings can be generalized to other ecosystems over an extended period, soil priming could make considerable contributions to the positive feedbacks between the terrestrial C cycle and climate warming." And then even stronger in the conclusions: "First, since warming has accelerated 335 positive soil priming, more soil C would be lost to the atmosphere under future climate change 336 scenarios if warming-enhanced plant biomass occurs. Consequently, the detrimental effects of 337 climate warming on temperate grassland ecosystems could be more severe than those expected 338 without considering soil priming 23." That is a really strong statement: increased priming means increased soil C loss. But it really doesn't deal with the point raised in my earlier review that if the authors contend their data contradict this meta-analysis published in Nature Communications: <https://www.nature.com/articles/s41467-018-05667-7> then they need to be clearer about addressing the controversy. It's important.

I believe the analysis contradicts the conclusion from this meta-analysis, but it's actually not clear in the paper. Given the importance of the carbon cycle feedback for the relevance of this paper, I think this basic point should be crystal clear. Specifically, the authors should state whether their analysis supports priming as a source or sink of soil C, address the arguments in the meta-analysis cited above, and clarify their conclusions and implications accordingly.

There are still places where I find the statements of evidence suggest causality when the evidence is too weak to support causal statements. I raised this issue in the first round of reviews focusing on this point about priming and net soil c, which remains ambiguous to me as described above. There are some other places the paper tends to overstate the strength of the evidence. For example, a section heading in the paper reads "Warming enhanced the priming effect by regulating active bacterial communities." Readers will understand this to be a statement about causality. I think this should be changed to a statement about correlation or association, given the nature of the test. Lines 192-194 are another strong claim of causality "soil C decomposition was strongly affected by the active functional genes" but the evidence is correlative, partial R² values which show association. I don't see any evidence based on a strong inference test for this claim, and it is possible that, for example, the temperature responsiveness of organisms is the key driver, among any other trait that might respond to the treatment.

The argument in lines 212-214 is weak. "The soils in our study is nutrient/N poor" is qualitative and not specific enough. I don't see any evidence that N limits microbial growth in these soils.

In the conclusions "experimental warming can regulate soil priming via altering the active microbial community's functional structure." The evidence for this is correlative and qualitative. I'm not convinced.

Note: these points about the strength of inference don't mean that the authors are necessarily wrong. It's just that the data don't support the claims as strongly as presented in the manuscript.

Line 187-190. I don't understand the logic of this sentence. How can bacterial community composition not affect bacterial abundance, when bacterial abundance is what defines bacterial community composition? This same circularity applies to phylogenetic diversity and functional gene abundance. Is there an interesting claim of causality here, too, and what is it?

For clarity, I suggest the authors avoid statements like "this result is more consistent with the prediction described in Extended Data Fig. 11a rather than Extended Data Fig. 11e." It isn't obvious to me what the references to these Extended Data figures means in terms of the substance of the argument. I think it needs to be presented here on its own merits, even if that description needs to be brief. Same with 218-219. On point iii of this argument, if N is so limiting in this system, why did plant growth not increase with higher SR and N availability? I do not mean to understate the challenge here, which is severe: how to disentangle extremely complex set of mechanisms in a single-factor experiment like this one, when such analyses often come down to arguments from correlation. That is the case here, and I think the authors need to be more up front about that.

Minor point: in figure 5, the legend describes "¹²C" CO₂ emissions. I believe the authors just mean CO₂ emissions, not specifically ¹²C, because natural abundance CO₂ includes a mix of both. This shorthand notation of using the common isotope to indicate the natural abundance treatment is rampant in the SIP literature, but it's incorrect and confusing, especially for those who work with natural abundance isotopes.

Line 320-322: "These results are also consistent with our previous analyses showing that microorganisms regulate soil carbon dynamics through three primary feedback mechanisms 7,15." What three mechanisms? This conclusion seems ancillary and should probably be dropped unless it is developed more clearly.

Reviewer #2:

Remarks to the Author:

The authors have addressed all my previous comments adequately. Such efforts were highly appreciated.

Reviewer #3:

Remarks to the Author:

The authors addressed the points I raised very thoroughly. I have no further remarks with regards to the microbial community analysis.

Responses to Comments from the Reviewers

(Manuscript # NCOMMS-23-19532A)

Reviewer #1 (Remarks to the Author):

A1. *I appreciate the authors' efforts in revising the manuscript and in the additional modeling. This part of the paper is improved. I wish there could be a more rigorous sensitivity analysis and model comparison, to probe more deeply the role of priming in processes involved in net C storage, and to explore the implications of the findings for long-term C cycling. Perhaps here, can the authors clarify whether the reduced parameter uncertainty could be a result of more parameters (and thus the potential for an over-fitted model?)*

Response:

We are grateful for the positive feedback on the revisions and additional modeling efforts, as well as for the suggestions to enhance the rigor of our analysis. The following are our responses to the concerns raised:

A), Regarding the need for a more rigorous sensitivity analysis, we have conducted a sensitivity analysis to evaluate the influence of parameters on simulated soil and microbial C pool sizes. The outcomes of this sensitivity analysis have been incorporated into Extended Data Figure 16a, with an in-depth discussion provided in Supplementary Text E. This discussion details the impact of various parameters on the simulated variables, as outlined in “For the current MEND application in field simulations, we conducted additional sensitivity analyses using the MOPSA method for a greater number of calibrated parameters and targeted variables, as shown in Extended Data Fig. 16a. The results indicated that the simulated variables are sensitive to different combinations of parameters; for instance, soil organic matter (SOM) was most sensitive to changes in Q_{max} (maximum sorption capacity), while microbial biomass carbon (MBC) was mostly sensitive to changes in Y_g . By averaging the sensitivity index (SI) ranks for targeted variables, Y_g , V_g , and KD emerged as the top three influential parameters, whereas fD and gD were the least influential among the 14 parameters selected during the calibration of field-MEND.” (Lines 154-163, Supplementary Text E). Additionally, the sensitivity ranking of parameters informed the order of parameter addition in the model comparison for impact of model complexity. This corresponding procedure is updated in the **Methods** section in **Lines 740-742** of the revised manuscript.

Extended Data Fig. 16 | MEND parameter sensitivity analysis and the impact of parameter number on model performance. a. The heatmap represents the sensitivity index for each parameter's effect on simulated variables in the MEND model. The sensitivity index is defined as the median of the discrepancies between acceptable and unacceptable parameter samples, as identified by the Multi-Objective Parameter Sensitivity Analysis (MOPSA) method. The terms ENZ^{SOM}, ENZ^{POMO}, and ENZ^{POMH} denote the sum of enzymes involved in C degradation and the enzyme pools EPO, EPH within MEND, respectively. The descriptions of variables and parameters are available in Supplementary Tables 8 and 11. b. This panel illustrates the influence of model complexity on training and test errors. The dataset, which includes variables such as Rh and gene abundance, was partitioned into a training set (the first 6 years) for model calibration and a test set (the subsequent 2 years) to evaluate model generalization. Model complexity is represented by calibrating different numbers of parameters, chosen based on their sensitivity index rankings from among 14 parameters in the field-MEND model. For each complexity level, three distinct parameter sets were evaluated. An increase in test error coupled with a decrease in training error, following the addition of parameters, could indicate overfitting to the training data²⁹. Conversely, a trend of decreasing test error with an increasing number of calibrated parameters may suggest that the model is appropriately complex and not overfitted. Prediction errors are quantified as 1-R² when comparing simulated and observed Rh, with error bars indicating the standard error from the three parameter sets at each complexity level. c. The effect of model complexity on parameter uncertainty is quantified using the Coefficient of Variation (CV). The CV for each complexity level is the average across all calibrated parameters.

(B), To clarify the role of priming in processes involved in net C storage, we have introduced a new simulation experiment that estimates the contributions of replenishment and priming loss, as well as the net change following litter/plant addition for both laboratory and field conditions. This approach draws upon the strategy from the referenced study¹ as suggested by the reviewer. Our simulation indicated that priming and replenishment together lead to a net increase in soil C within both laboratory and field modeling. This finding is now documented as *“In both laboratory and field experiments, the combinations of priming and replenishment yielded net increases in soil carbon (Extended Data Fig. 18) under both warming and control, which is consistent with the results from a previous data-model synthesis study based on 84 priming experiments⁷⁵.”* (Extended Data Figure 18, Lines 330-333 of the revised manuscript). Furthermore, we have expanded the discussion to consider how warming treatments might affect net soil C storage in such a system, shown as *“Our results also revealed that the experimental warming may decrease the net soil C gain by reducing replenishment by 1.7 % (scaled to percent of annual plant C input) and increasing the priming effect by 9.1 % under the field condition.”* (Extended Data Figure 18, Lines 333-336 of the revised manuscript). The methodology employed for these estimations is described in the Methods section (Lines 809-817 of the revised manuscript).

Extended Data Fig. 18 / Annual SOC change induced by replenishment and priming, and the consequent net SOC change simulated by lab-MEND and field-MEND. a. Estimation of SOC changes under lab incubation conditions. “Replenishment” refers to the amount of new (added) C remaining in soil C pools after microbial respiration within a year of simulation. The “priming effect” is the difference in C loss from native SOC between the substrate addition treatment and the control. The net effect of litter addition on SOC change is the difference between replenishment and priming. The change in SOC induced by each process was scaled to the initially added litter C amount. b. Estimation of SOC changes under field conditions in 2016. For the field condition, litter addition is replaced by plant carbon input. The priming effect is the difference in C loss from native SOC between the normal condition (with plant carbon input) and an assumed condition (without plant carbon input for a whole year).

(C), In terms of the potential for model overfitting due to an increase in parameters, we have included a model comparison analysis that tests the effects of parameter count on model generalization and parameter uncertainty (Extended Data Figures 16b and c). We compared the MEND model's performance at varying levels of complexity to determine how the number of

parameters affects both model generalization and parameter uncertainty. Our analysis confirms that an increase in parameters does not lead to overfitting in our case. However, it does initially elevate parameter uncertainty, which subsequently decreases. We selected the number of parameters that offer the lowest uncertainty while maintaining optimal model fit. These results and the corresponding discussion of the effects of parameter numbers on model generalization and parameter uncertainty have been added to **Lines 282-300 of the revised version**, shown as “In general, simple models with very few parameters may underfit the experimental data, failing to capture the variations in datasets containing multiple observed variables. Conversely, complex models with an excess of parameters may overfit the data, leading to poor generalization⁷³. To test the impacts of model complexity on model generalization, we compared the test error (unexplained variance of the test dataset) for field-MEND models with an increasing number of parameters calibrated with the training dataset (Supplementary Text E and Extended Data Fig. 16b). Increasing model parameters may lead to reduced unexplained variance in the training set. However, if test error rises simultaneously, it suggests overfitting—the model is overly complex, fitting too closely to the training set and struggling to predict new data accurately. Our results showed that the test error decreased with an increasing number of parameters (Extended Data Fig. 16b), suggesting improved model prediction for test data without overfitting. Nevertheless, in addition to the number of parameters, the consistently higher test error compared to the training error suggests the need for further enhancements in the model's structure to improve its generalization capabilities⁷⁴. Furthermore, the calibration of more parameters initially increased parameter uncertainty and later decreased it (Supplementary Text E and Extended Data Fig. 16c), which indicated that the 14-parameter MEND had the least parameter uncertainty among the models with the lowest test error. Overall, the current parameter selection improves model generalization, alleviating the overfitting problem without a notable increase in parameter uncertainty. The relevant methodology has also been detailed in the **Methods section in Lines 736-746 of the revised manuscript**.

A2. *I'm concerned readers will be confused about what these findings say about the effect of priming on net soil C accumulation or loss. Are the authors arguing that warming-induced priming is a net source of C loss? Does that statement account for the contributions of microbial growth and death during priming to C storage? In the response letter, the authors note that “we only used the model to predict C loss induced by the priming effect but not the net soil C changes in our original manuscript.” But the conclusions still lean on the priming as a part of the climate-carbon positive feedback:*

“Model simulations from 2010 to 2016 indicated a notable increase in soil C decomposition under warming, with a 7% rise in priming-induced CO₂ emissions. If our findings can be generalized to other ecosystems over an extended period, soil priming could make considerable contributions to the positive feedbacks between the terrestrial C cycle and climate warming.” And then even stronger in the conclusions: “First, since warming has accelerated 335 positive soil priming, more soil C would be lost to the atmosphere under future climate change 336 scenarios if warming-enhanced plant biomass occurs. Consequently, the detrimental effects of 337 climate warming on temperate grassland ecosystems could be more severe than those expected 338 without considering soil priming 23. “ That is a really strong statement: increased

priming means increased soil C loss. But it really doesn't deal with the point raised in my earlier review that if the authors contend their data contradict this meta-analysis published in Nature Communications:

<https://www.nature.com/articles/s41467-018-05667-7> then they need to be clearer about addressing the controversy. It's important.

I believe the analysis contradicts the conclusion from this meta-analysis, but it's actually not clear in the paper. Given the importance of the carbon cycle feedback for the relevance of this paper, I think this basic point should be crystal clear. Specifically, the authors should state whether their analysis supports priming as a source or sink of soil C, address the arguments in the meta-analysis cited above, and clarify their conclusions and implications accordingly.

Response: We appreciate the insightful questions and the opportunity to clarify our intentions.

1), Regarding the first question on the effect of priming on net soil C dynamics, we would like to clarify that our manuscript does not posit warming-induced priming as a definitive net source of carbon loss for the entire system. Our primary focus was to highlight the potential for increased soil CO₂ emissions as a consequence of intensified positive soil priming effects under warming conditions. We are sorry for any confusion that may have arisen from our use of the term "loss" and "positive feedback" and acknowledge that it might have been misleading. Our intention was not to suggest a net soil carbon loss or gain, but rather to emphasize the increased CO₂ emissions process.

2), For the second question, as we stated above, the conclusion was drawn only based on priming-induced CO₂ emissions, and therefore the conclusion does not account for priming-associated replenishment processes. To address these concerns:

Firstly, based on the methods in Liang et al. 2018¹, we have estimated priming, replenishment, and net carbon change for our lab incubation and field settings via MEND model. The results (Extended Data Fig. 18) show consistent pattern with Liang et al. 2018 (Figure 2)¹, where both replenishment and priming dynamics were considered. The synergistic effect in both our lab incubation and field modeling indicates a net gain in soil C. As suggested, we have inserted a clear statement to this effect in the revised manuscript at **Lines 330-333**, shown as "In both laboratory and field experiments, the combinations of priming and replenishment yielded net increases in soil carbon (Extended Data Fig. 18) under both warming and control, which is consistent with the results from a previous data-model synthesis study based on 84 priming experiments⁷⁵." The methodology employed for these estimations is described in the Methods section (**Lines 809-817 of the revised manuscript**).

Secondly, we have revised the conclusion to better reflect our findings: *“Firstly, our results revealed that warming intensified positive soil priming effects, leading to increased soil CO₂ emissions to the atmosphere, especially if warming coincides with enhanced plant biomass production. Therefore, the effects of climate warming on temperate grassland ecosystems could be more pronounced than previously estimated when the dynamic of soil priming is factored in. This nuanced understanding underscores the complexity of soil carbon responses to climate warming and the need for integrated models that capture the interplay between soil carbon dynamics and ecosystem processes”*²³.” (Lines 362-369 of the revised version). Furthermore, we also changed the term “priming-induced C loss” to *“priming-induced CO₂ emissions”* (Lines 42 of the revised version), and delete the term “positive feedbacks” from the abstract in our revised version to prevent any potential misinterpretation, shown as *“If our findings can be generalized to other ecosystems over an extended period, soil priming could play a substantial role in influencing the feedbacks between the terrestrial C cycles and climate warming.”* in Lines 42-44 of the revised version.

In summary, our results are consistent with Liang et al. 2018¹. As suggested, we have stated clearly that the magnitude of replenishment is greater than that of priming, resulting in a net increase in SOC in both control and warmed soils.

A3. *There are still places where I find the statements of evidence suggest causality when the evidence is too weak to support causal statements. I raised this issue in the first round of reviews focusing on this point about priming and net soil c, which remains ambiguous to me as described above. There are some other places the paper tends to overstate the strength of the evidence. For example, a section heading in the paper reads “Warming enhanced the priming effect by regulating active bacterial communities.” Readers will understand this to be a statement about causality. I think this should be changed to a statement about correlation or association, given the nature of the test. Lines 192-194 are another strong claim of causality “soil C decomposition*

was strongly affected by the active functional genes” but the evidence is correlative, partial R² values which show association. I don’t see any evidence based on a strong inference test for this claim, and it is possible that, for example, the temperature responsiveness of organisms is the key driver, among any other trait that might respond to the treatment.

The argument in lines 212-214 is weak. “The soils in our study is nutrient/N poor” is qualitative and not specific enough. I don’t see any evidence that N limits microbial growth in these soils.

In the conclusions “experimental warming can regulate soil priming via altering the active microbial community’s functional structure.” The evidence for this is correlative and qualitative. I’m not convinced.

Note: these points about the strength of inference don’t mean that the authors are necessarily wrong. It’s just that the data don’t support the claims as strongly as presented in the manuscript.

Response: Thanks for pointing this out. We agree that the “correlation analysis” results cannot be claimed as strongly as presented in the manuscript. As suggested, we’ve softened the tone in the PLS modeling analysis section (Lines 180-197 of the original manuscript) to primarily discuss the correlations and associations in our updated version. Specifically,

- (i) We have revised the section heading from “Warming enhanced the priming effect by regulating active bacterial communities” to “Priming effect is associated with active bacterial communities under warming” to more accurately reflect the nature of our findings (**Line 180 of the revised version**).
- (ii) The statements in lines 183-195 in original manuscript have been modified to “The analysis found a notable association between soil temperature and active bacterial communities in terms of their abundance (partial R² = 0.46, p < 0.01, based on the PLS model), phylogenetic diversity (partial R² = 0.30, p < 0.01), and functional genes related to C decomposition (partial R² = 0.25, p < 0.050). Mineral N also showed a significant association with soil temperature (partial R² = 0.16, p < 0.05) and active bacterial abundance (partial R² = 0.20, p < 0.01). As anticipated, the composition of active bacterial communities were correlated with total bacterial abundance (partial R² = 0.32, p < 0.01), phylogenetic diversity (partial R² = 0.39, p < 0.01), functional gene abundance (partial R² = 0.47, p < 0.01), carbohydrate (partial R² = 0.34, p < 0.01), condensed aromatics (partial R² = 0.46, p < 0.05), and tannins abundance (partial R² = 0.34, p < 0.05). Additionally, soil C decomposition had a notable association with active functional genes related to vanillin-lignin decomposition (partial R² = 0.48, p < 0.01), and exhibited a significant correlation with the mineral N (partial R² = 0.42, p < 0.05) in the soil.” To highlight the association rather than causation (**Lines 183-195 of the revised version**).
- (iii) We have revised the conclusion sentence to “These findings suggest that priming effects under warming either directly or indirectly related to the changes in active microbial community functional genes and N availability (Supplementary text D).” (**Lines 195-197**

of the revised version) to only give potential suggesting findings for readers without overstating the evidence.

In reference to lines 212-214 of our original manuscript, we agree that the statement “The soils in our study are nutrient/N poor” is inaccurate. Our primary intent was to emphasize that our observations (as shown in Figure 4a) align more closely with the stoichiometric hypothesis scenarios. Specifically, when the C/N ratio surpasses the microbial optimum, an increase in available N brings the C/N ratio nearer to this optimal value. This in turn augments microbial activity, leading to enhanced C decomposition and a stronger priming effect, as depicted in Extended Data Fig. 11a. We have realized that our initial phrasing might have been misinterpreted as an assertion about the general nutrient status of our soils. Additionally, using the terms “Low-N availability” and “High-N availability” in Extended Fig. 11 to outline the stoichiometric decomposition scenarios was imprecise and could potentially confuse readers. To rectify this, we have:

- (i) Updated the terms to ‘C/N > C/N optimal’ and ‘C/N < C/N optimal’ in Extended Fig. 11 to more accurately reflect the stoichiometric decomposition scenarios.
- (ii) In light of the following Comment A5, we’ve removed the aforementioned argument from our revised version and have restructured this section of the discussion to “*Our results indicate that (i) Mineral N increased with native soil respiration stimulated by the addition of fresh C (Fig. 4b; $R^2 = 0.51$, $p < 0.050$); (ii) Plant biomass increased with decreased native soil respiration stimulated by the addition of fresh C (Fig. 4c; $R^2 = 0.60$, $p = 0.062$). These observations align more closely with the scenarios predicted by the “stoichiometric decomposition” hypothesis, especially for the C/N ratio exceeds the microbial optimal scenarios (Extended Data Fig. 11a-11b). Furthermore, the stimulation of potential r-strategists, such as α -Proteobacteria and Bacilli, by the addition of fresh C (Supplementary Table 3, Supplementary Table 5, and Supplementary text D) is in line with previous observations that r-strategists’ dominance elucidates the enhanced SOM decomposition based on the “stoichiometric decomposition” hypothesis³³. Therefore, although the roles of “microbial N mining” could be not completely ruled out, our data are more consistent with the “stoichiometric decomposition” hypothesis.*” for clarity (**Lines 217-227 of the revised version**).

A4. Line 187-190. I don’t understand the logic of this sentence. How can bacterial community composition not affect bacterial abundance, when bacterial abundance is what defines bacterial community composition? This same circularity applies to phylogenetic diversity and functional gene abundance. Is there an interesting claim of causality here, too, and what is it?

Response: Thank you for pointing this out. We agree with that, in a majority of instances, alterations in bacterial community composition would have effects on total bacterial abundance, phylogenetic diversity, and functional gene abundance. However, there are specific scenarios where changes in community composition might not significantly change the other parameters:

1. **Total Bacterial Abundance:** In general, total biomass is more influenced by the overall resource level than by composition changes. A change in composition doesn't necessarily result in a change in total biomass. 2. **Phylogenetic Diversity (PD):** PD typically relates to niche selection. If niche diversity remains consistent, composition changes (e.g., due to ecological drift) might not significantly alter PD. For instance, a transition from one species to another closely related one might not substantially change the overall phylogenetic diversity, given that both species occupy nearby branches on the phylogenetic tree. 3. **Functional Gene Abundance:** Generally, changes in composition should alter community functional profiles, since different species often possess different functional attributes. However, functional redundancy is a common phenomenon in microbial communities. Therefore, a shift in species might not necessarily result in a change in the overall abundance of specific functional genes.

While these scenarios are theoretically possible, they might not always be the case in real-world ecosystems. The interconnected nature of microbial communities often means that a change in one aspect (like composition) can lead to changes in others (like functional gene abundance).

Regarding lines 187-190 of the original manuscript, our intention was not to present groundbreaking findings but to ensure a comprehensive depiction of our observations related to warming's impact on the priming effect via active bacterial communities. To provide clarity and address potential ambiguities:

- We've adjusted the term "the bacterial abundance" to "total bacterial abundance" to clearly convey that it represents the cumulative number, not individual ASV abundance (**Line 189 of the revised version**).

- We've prefaced the sentence with "As anticipated" to signal to readers that these observations, while expected, are integral for the comprehensive understanding of our study's context (**Line 188 of the revised version**).

A5. *For clarity, I suggest the authors avoid statements like "this result is more consistent with the prediction described in Extended Data Fig. 11a rather than Extended Data Fig. 11e." It isn't obvious to me what the references to these Extended Data figures means in terms of the substance of the argument. I think it needs to be presented here on its own merits, even if that description needs to be brief. Same with 218-219. On point iii of this argument, if N is so limiting in this system, why did plant growth not increase with higher SR and N availability? I do not mean to understate the challenge here, which is severe: how to disentangle extremely complex set of mechanisms in a single-factor experiment like this one, when such analyses often come down to arguments from correlation. That is the case here, and I think the authors need to be more up front about that.*

Response: Thank you for the constructive feedback. We recognize the ambiguity in lines 213-219 of the original manuscript and appreciate your suggestion for greater clarity. Additionally, our assertion on lines 214-215 of the original manuscript, stating “The soils in our study are nutrient/N poor,” was inaccurately phrased and may have led to confusion. We apologize for this oversight.

To clarify, as detailed in our Response to Comment A3, our main objective was to emphasize that our findings (Figures 4a and 4b) align more closely with the scenarios of the stoichiometric hypothesis, that is when the C/N ratio exceeds the microbial optimum, an uptick in available N tends to bring this ratio closer to the ideal value. Consequently, this catalyzes microbial activity, resulting in enhanced carbon decomposition and a pronounced priming effect. This mechanism aligns with what is portrayed in Extended Data Fig. 11a.

It’s essential to note that our intention isn’t to provide a blank statement about the nutrient status of our soils or to categorically determine if the nutrient is limiting or abundant for plants and microbes. Rather, our focus lies in juxtaposing our two observations: firstly, “Mineral N increased with native soil respiration upon the introduction of fresh C,” and secondly, “Plant biomass surged when native soil respiration decreased due to the infusion of fresh C.” Our aim is to discern which theory, between the “stoichiometric hypothesis” and the “microbial N mining,” best encapsulates our above observations.

To address these concerns and obviate potential confusion, we have undertaken two steps. Firstly, recognizing that using the terms “Low-N availability” and “High-N availability” in Extended Fig. 11 was imprecise and could mislead readers, we’ve updated them to ‘C/N > C/N optimal’ and ‘C/N < C/N optimal’ in Extended Fig. 11 to more appropriately represent the stoichiometric decomposition scenarios. Secondly, we have carefully revised and rephrased the discussion on priming mechanisms in relation to the “stoichiometric hypothesis” and the “microbial N mining”. This revised section can now be found as “Theoretically, two prominent competing hypotheses have been proposed to elucidate the soil priming effect: the “stoichiometric decomposition” hypothesis and the “microbial N mining” hypothesis^{29,33} (Extended Data Fig. 11). The “stoichiometric decomposition” hypothesis assumes that microbial activities, including decomposition and respiration, would be highest when the stoichiometry of substrates matches that of microbial demands^{33,63} (Extended Data Fig. 11a-11d). Specifically, when the C/N ratio exceeds the microbial optimal, an increase in available N draws the C/N ratio closer to this optimal. This enhances microbial activities, thus increasing C decomposition and the priming effect (Extended Data Fig. 11a). Conversely, an increase in fresh C input shifts the C/N ratio away from the optimal, which could induce N deficiency, weaken microbial activity, and lead to a reduction in C decomposition and the priming effect (Extended Data Fig. 11b). In contrast, the “microbial N mining” hypothesis posits that microorganisms can use labile C as an energy source to decompose the native SOC for additional nutrient/nitrogen (N)^{26,64} (Extended Data Fig. 11e-11i). Our results indicate that (i) Mineral N increased with native soil respiration stimulated by the addition of fresh C (Fig. 4b; $R^2 = 0.51$, $p < 0.050$); (ii) Plant biomass increased with decreased native soil respiration stimulated by the addition of fresh C (Fig. 4c; $R^2 = 0.60$, $p = 0.062$). These observations align more closely with the scenarios

predicted by the “stoichiometric decomposition” hypothesis, especially for the C/N ratio exceeds the microbial optimal scenarios (Extended Data Fig. 11a-11b). Furthermore, the stimulation of potential r-strategists, such as α -Proteobacteria and Bacilli, by the addition of fresh C (Supplementary Table 3, Supplementary Table 5, and Supplementary text D) is in line with previous observations that r-strategists' dominance elucidates the enhanced SOM decomposition based on the “stoichiometric decomposition” hypothesis³³. Therefore, although the roles of “microbial N mining” could be not completely ruled out, our data are more consistent with the “stoichiometric decomposition” hypothesis.” in **Lines 204-227 of the updated manuscript.**

A6. *Minor point: in figure 5, the legend describes “12-C” CO₂ emissions. I believe the authors just mean CO₂ emissions, not specifically 12C, because natural abundance CO₂ includes a mix of both. This shorthand notation of using the common isotope to indicate the natural abundance treatment is rampant in the SIP literature, but it's incorrect and confusing, especially for those who work with natural abundance isotopes.*

Response: Thank you for pointing this out. Indeed, we intended to describe CO₂ emissions (12-C + 13-C CO₂) other than 12-C CO₂ emissions. As suggested, we revised the legend to “a, CO₂ respired from the soil without straw addition”.

A7. *Line 320-322: "These results are also consistent with our previous analyses showing that microorganisms regulate soil carbon dynamics through three primary feedback mechanisms 7,15." What three mechanisms? This conclusion seems ancillary and should probably be dropped unless it is developed more clearly.*

Response: Thank you for the comment. As suggested, we have dropped this conclusion.

Reviewer #2 (Remarks to the Author):

B1. *The authors have addressed all my previous comments adequately. Such efforts were highly appreciated.*

Response: Thank you for acknowledging our efforts in addressing your comments. We are grateful for your guidance and positive feedback throughout this review process.

Reviewer #3 (Remarks to the Author):

C1. The authors addressed the points I raised very thoroughly. I have no further remarks with regards to the microbial community analysis.

Response: Thank you for acknowledging our detailed work on the microbial community analysis and we appreciate your valuable input throughout this process.

References:

- 1 Liang, J. *et al.* More replenishment than priming loss of soil organic carbon with additional carbon input. *Nature communications* **9**, 3175 (2018).

Reviewers' Comments:

Reviewer #1:

Remarks to the Author:

I am satisfied that the revision has addressed my criticisms.

Responses to Comments from the Reviewers

(Manuscript # NCOMMS-23-19532B)

Reviewer #1 (Remarks to the Author):

A1. I am satisfied that the revision has addressed my criticisms.

Response: Thank you for acknowledging our efforts in addressing your criticisms. We are grateful for your guidance and positive feedback throughout this review process.